# Data-Dependent Generalization Bounds for Neural Networks with ReLU

**Harsh Pandey**  *bokharsh@gmail.com*
*Department of Computer Science*
*IIT Delhi*
*New Delhi, India*

**Amitabha Bagchi**  *bagchi@cse.iitd.ac.in*
*Department of Computer Science*
*IIT Delhi*
*New Delhi, India*

**Srikanta Bedathur**  *srikanta@cse.iitd.ac.in*
*Department of Computer Science*
*IIT Delhi*
*New Delhi, India*

**Arindam Bhattacharya**  *arindambhattacharya@protonmail.com*
*Department of Computer Science*
*IIT Delhi*
*New Delhi, India*

**Reviewed on OpenReview:** *https://openreview.net/forum?id=mH6TelHVKD*

## Abstract

We try to establish that one of the correct data-dependent quantities to look at while trying to prove generalization bounds, even for overparameterized neural networks, are the gradients encountered by stochastic gradient descent while training the model. If these are small, then the model generalizes. To make this conclusion rigorous, we weaken the notion of uniform stability of a learning algorithm in a probabilistic way by positing the notion of almost sure (a.s.) support stability and showing that algorithms that have this form of stability have generalization error tending to 0 as the training set size increases. Further, we show that for Stochastic Gradient Descent to be a.s. support stable we only need the loss function to be a.s. locally Lipschitz and locally Smooth at the training points, thereby showing low generalization error with weaker conditions than have been used in the literature. We then show that Neural Networks with ReLU activation and a doubly differentiable loss function possess these properties. Our notion of stability is the first data-dependent notion to be able to show good generalization bounds for non-convex functions with learning rates strictly slower than $1/t$ at the $t$-th step. Finally, we present experimental evidence to validate our theoretical results.

## 1 Introduction

Deep neural networks are known to perform well on unseen data (test data), c.f. e.g. Jin et al. (2020)), but theoretical explanations of this behaviour are still unsatisfactory. Under the assumption that the error on the training set (empirical error) is low, studying the gap between the empirical error and the population error is one route to investigating why this performance is good. In this paper we look at the gap between the population error (risk) and empirical error (empirical risk) 1. Following works like Bousquet & Elisseeff (2002), we use the term *generalization error* for this quantity. Chatterjee & Zielinski (2022) articulated the

| Paper | Number of Epochs | Step Size | Neural Network Type | Key Assumptions |
|---|---|---|---|---|
| Hardt et al. (2016) | $O(m^c), c > 1/2$ | $O(1/t)$ | No restrictions | No data-dependence. |
| Kuzborskij & Lampert (2018) | 1 epoch | $O(1/t)$ | No restrictions | Bounded Hessian |
| Lei & Ying (2020) | $O(1)$ | $O(1/t)$ | No restrictions | Strongly convex objective but non convex loss function |
| Charles & Papailiopoulos (2018) | $O(m)$ | $O(1)$ | 1-layered networks with leaky ReLU or linear | PL and QG growth conditions |
| Lei et al. (2022) | $O(m)$ | $O(1)$ | 1-layered networks with smooth activation functions | Smooth loss function, Bound in expectation, lower bound on number of parameters $n > m^{\frac{3}{(\alpha+1)}}, \alpha > 0$ |
| Our Paper | $O(\log m)$ | $O\left(1/t^{1-\frac{c}{\rho(\tau,m)}}\right), c \in (0,1)$ | No restrictions | Bounded Spectral Complexity |

Table 1: Recent related works addressing the question of generalization error and stability of neural networks in comparison to the results in this paper.

main question as follows: *why (or when) do neural networks generalize well when they have sufficient capacity to memorize their training set?* Although a number of formalisms have been used in an attempt to derive theoretical bounds on the generalization error, e.g., VC dimension (Vapnik, 1998), Rademacher complexity (Bartlett & Mendelson, 2003) and uniform stability (Bousquet & Elisseeff, 2002) but, as Zhang et al. (2017) showed, all of these fail to resolve the conundrum thrown up by overly parameterized deep neural networks. One clear failing identified in Zhang et al. (2017) was that many of these notions were data-independent. A simple counterexample provided by Zhang et al. (2017) clearly established that a data-independent notion was bound to fail to distinguish between data distributions on which deep NNs will generalize well and those on which they will not. Further, the paper also raised doubts on the possibility of proving so-called uniform convergence bounds for generalization error, i.e., bounds that were independent of the the size of the training data. These doubts were concretized by Nagarajan & Kolter (2019) who constructed examples for which the best possible algorithms were shown to not have uniform convergence bounds. We do not directly address the issue of uniform convergence in this paper, instead focusing on the question that Chatterjee & Zielinski (2022) formulated as follows: *For a neural network, is there a property of the dataset that controls the generalization error (assuming the size of the training set, architecture, learning rate, etc are held fixed)?* We give an affirmative answer to this question in one direction: We identify the data-dependent quantities, namely SMSTrG, WCTrG and TeG and show that if these are bounded, we can guarantee the generalization of neural networks. We show that the bound on these quantities depends on the difference between the initial value and optimal value of the loss function and also do experiments to validate our results. Our techniques are able to rigorously handle nonlinearities like ReLU and work for non-convex loss functions, and this holds for classification case. We also allow for a learning rate that is asymptotically strictly slower than $\theta(1/t)$ at the $t$-th step of SGD. All this holds for any bounded value loss function, which is twice differentiable.

| Name | Shorthand | Mathematical Notation |
|---|---|---|
| Worst Case Training Gradient | WCTrG | $L_S$ |
| Second Moment of Step Training Gradient | SMSTrG | $\sigma_S$ |
| Test Gradient | TeG | $L_g$ |
| Training Smoothness Constant | - - | $K_S$ |

Table 2: These are the important random variables which are used in the paper. All these are defined in Section 4.1.1.

Our work is within the theoretical paradigm of stability. We asked the question, *Is there an appropriate version of stability that is flexible enough to incorporate dataset properties and can also adapt to most neural networks?* In a partial answer to this question, we introduce a notion called *almost sure (a.s.) support stability* which is a data-dependent probabilistic weakening of uniform stability. Following the suggestions made by Zhang et al. (2017), data-dependent notions of stability were defined in (Kuzborskij & Lampert, 2018, Definition 2) and (Lei & Ying, 2020, Definition 4) as well. However, a.s. support stability is a more useful notion on three counts: it can handle SGD learning rates that are strictly slower than $\theta(1/t)$, its initial learning rate is much higher, and, while these past works bound generalization error in expectation, a.s. support stability can be used to show high probability bounds on generalization error. But, over and above these technical benefits, our main contribution here is the identification of the data-dependent quantities as a key indicator of generalization. Which in turn connects the generalization of neural networks to the difference of initial and optimal loss values. A brief description of recent related works are summarized in table 1.

Earlier works for showing generalization (like Kuzborskij & Lampert (2018)) have a global Lipschitz constant and show generalization using some other parameter. More recent works like Lei & Ying (2020) try to completely remove the role of Lipschitz constants by taking some other assumptions on the structure or on the loss function. We argue that the gradients are the correct quantities to look at for generalization.

We show generalization bounds for neural networks via two paths. One path uses Worst Case Training Gradient (WCTrG) which is the the worst-case gradient across all steps of SGD and *Test Gradient* (TeG) which is a gradient computed at the final parameter vector computed by the training. The other path uses the *Second Moment of Step Training Gradient* (SMSTrG) which is related to the second moment of the gradients encountered during training and TeG. Using some results from Bottou et al. (2018) we show a bound on SMSTrG in terms of the difference between the initial loss and optimal loss values. This directly bounds the generalization error of neural networks in terms of this difference and makes this bound more usable. We show examples of cases that take advantage of our bounds. We perform experiments to validate the results and also empirically show that for random label case, WCTrG grows unboundedly, and so we can't guarantee generalization in this case, which is as expected.

We note that although we can say that when the data-dependent quantities are small, our results guarantee good generalization performance, we do not establish that this condition is necessary.

In particular, our contributions are:

• In Section 3 we define a new notion of stability called *a.s. support stability* and show in Theorem 3.2 that algorithms with a.s. support stability $o(1/\log^2 m)$ have generalization error tending to 0 as $m \to \infty$ where $m$ is the size of the training set.

• In Section 4 we first define the data-dependent quantities. We run SGD for $\tau$ epochs with the slowest learning rate of $\alpha_0/t^{1-\rho(\tau,m)}$, where $\rho(\tau,m) = O(\log\log m/(\log\tau + \log m))$ for appropriate value of $\alpha_0$. For reasonable values of $m$ and $\tau$, this marks a significant slowing down of the learning rate from $\theta(1/t)$. We use two different ways to show a.s. support stability of SGD and we also show a bound on SMSTrG based on SGD properties.

– In Section 4.1 we define the data-dependent quantities and show their existence.

- In Section 4.2 we show a.s. support stability of SGD using WCTrG and TeG. We show this for learning rate of $\alpha_0/t^{1-\rho(\tau,m)}$, where $\rho(\tau,m) < 1$ for reasonable size of training set $(m)$ and epochs $(\tau)$ proportional to $\log(m)$.
- In Section 4.3 we show a.s. support stability of SGD using SMSTrG and TeG where $\tau = 1$. This also enjoys the small learning rate of $\alpha_0/t^{1-\rho(\tau,m)}$.
- In Section 4.4, we show a bound on SMSTrG based on properties of SGD and a very minor assumption (P1) as highlighted by Bottou et al. (2018). The main highlight of the bound on SMSTrG is that this takes advantage of the fact that even for non-convex optimization the gradients of loss function decrease as shown by Bottou et al. (2018).

- In Section 5, we combine results from Section 3 and 4 to show generalization error bound for neural networks. We also translate these bound to the neural network setting. And show practical examples highlighting the advantages of our data-dependent constants.

  - In Section 5.1 using Section 4.2, we bound WCTrG by the spectral property (Proposition 5.3) and show generalization bounds based on these spectral properties.
  - In Section 5.2 using Sectoin 4.3 and 4.4, we show generalization via SMSTrG and show bound on it which depends on the difference of initial loss and optimal loss values. The main advantage of this is that this is much more practical and easy to verify as compared to the spectral property.

- Then, in Section 6, we experimentally verify the results showing that the bounded condition holds and plot the generalization error. We also experimentally analyze the Test Gradient (TeG) for random labelling setting suggested by Zhang et al. (2018) and conclude the Test Gradient is actually not bounded and increases with the training set size. We relate this to the high variance of the loss function in random labelling case and hence provide an explanation of which this example cannot be proved incorrectly to generalize using our methods.

## 2 Related Work

Although NNs are known to generalize well in practice, many different theoretical approaches have been tried without satisfactorily explaining this phenomenon, c.f., Jin et al. (2020); Chatterjee & Zielinski (2022). We refer the reader to the work of Jin et al. (2020) which presents a concise taxonomy of these different theoretical approaches. Several works seek to understand what a good theory of generalization should look like, c.f. Kawaguchi et al. (2017); Chatterjee & Zielinski (2022). Our own work falls within the paradigm that seeks to use notions of algorithmic stability to bound generalization error that began with Vapnik & Chervonenkis (1974) but gathered steam with the publication of the work by Bousquet & Elisseeff (2002).

The applicability of the algorithmic stability paradigm to the study of generalization error in NNs was brought to light by Hardt et al. (2016), who showed that functions optimized via Stochastic Gradient Descent have the property of uniform stability defined by Bousquet & Elisseeff (2002), implying that NNs should also have this property. Subsequently, there was renewed interest in uniform stability, and a sequence of papers emerged using improved probabilistic tools to give better generalization bounds for uniformly stable algorithms, e.g., Feldman & Vondrak (2018; 2019a) and Bousquet et al. (2020). Some other works, e.g. Klochkov & Zhivotovskiy (2021), took this line forward by focussing on the relationship of uniform stability with the excess risk. However, the work of Zhang et al. (2017) complicated the picture by pointing out examples where the theory suggests the opposite of what happens in practice. This led to two different strands of research. In one thread, an attempt was made to either discover those cases where uniform stability fails (e.g. Charles & Papailiopoulos (2018)) or to show lower bounds on stability that ensure that uniform stability does not exist (e.g. Zhang et al. (2022)). The other strand of research, a category in which our work falls, focuses on weakening the notion of uniform stability, specifically by making it data-dependent, thereby following the suggestion made by Zhang et al. (2017). (Kuzborskij & Lampert, 2018, Definition 2) defined "on-average stability" which is weaker than our definition of a.s. support stability. Consequently, their definition leads to a weaker in-expectation bound on the generalization error where the expectation is over the training set as well as the random choices of the algorithm. Our Theorem 3.2, on the other hand, provides a sharp concentration bound on the choice of the training set. (Lei & Ying, 2020, Definition 4)

define an "on-average model stability" that requires the average replace-one error over all the training points to be bounded in expectation. While their smoothness requirements are less stringent, the problem is that their generalization results are all relative to the optimal choice of the weight vector, which implies a high generalization error in case of early stopping.

A key question in the study of generalization in NNs is that of the possibility of proving uniform convergence bounds for norm-bounded classes of NNs, a question that can be traced to Bartlett (1996). This question was answered negatively by Nagarajan & Kolter (2019) who showed that in general norm bounds on deep networks show non-uniform growth, i.e., they grow with the training set size. For further discussion on this question the reader is referred to Negrea et al. (2020a). Our current work is more focussed on identifying the relevant quantities that bound generalization error, postponing the question of the exact relationship of these quantities to the training set size to future study.

The role of the norm of gradients for bounding generalization error has been observed from a different angle by Negrea et al. (2020b) and also by Haghifam et al. (2020). Their work focuses on giving information-theoretic generalization bounds for Stochastic Gradient Langevin Dynamics (SGLD) and Langevin dynamics algorithm. Although their setting and methods is different from ours, the general flavour is similar.

## 3 Almost Sure (a.s.) Support Stability and Generalization

In this section, we present a weakening of the notion of uniform stability defined by (Bousquet & Elisseeff, 2002, Definition 6) and show that exponential concentration bounds on the generalization error can be proved for learning algorithms that have this weaker form of stability.

### 3.1 Terminology

Let $\mathcal{X}$ and $\mathcal{Y}$ be the input and output spaces respectively. We assume we have a training set $S \in \mathcal{Z}^m$ of size $m$ where each point is chosen independently at random from an unknown distribution $D$ over $\mathcal{Z} \subset \mathcal{X} \times \mathcal{Y}$. For $z = (x, y) \in \mathcal{Z}$ we will use the notation $x_z$ to denote $x$ and $y_z$ to denote $y$. Let $\mathcal{R}$ be the set of all finite strings on some finite alphabet, and let us call the elements of $\mathcal{R}$ *decision strings* and let us assume that there is some probability distribution $D_r$ according to which we will select $r$ randomly from $\mathcal{R}$. This random string abstracts the random choices of the algorithm. For example, in an NN trained with SGD it encapsulates the random initial parameter vector and the random permutation of the training set as seen by SGD. For an algorithm like Random Forest $r$ would abstract out the random points chosen to divide the space.

Further, let $\mathcal{F}$ be the set of all functions from $\mathcal{X}$ to $\mathcal{Y}$. In machine learning settings we typically compute a map from $\mathcal{Z}^m \times \mathcal{R}$ to $\mathcal{F}$. We will denote the function computed by this map as $A_{S,r}$. Since the choice of $S$ and $r$ are both random, $A_{S,r}$ is effectively a random function and can also be thought of as a randomized algorithm.

Given a constant $M > 0$, we assume that we are given a bounded *loss function* $\ell : \mathcal{Y} \times \mathcal{Y} \to [0, M]$. We define the *risk* of $A_{S,r}$ as

$$R(A_{S,r}) = \mathrm{E}_{z \sim D} \left[ \ell(A_{S,r}(x_z), y_z) \right],$$

where the expectation is over the random choice of point $z$ according to data distribution $D$. Note that the risk is a random variable since both $S$ and $r$ are randomly chosen. The *empirical risk* of $A_{S,r}$ is defined as

$$R_e(A_{S,r}) = \frac{1}{|S|} \sum_{z \in S} \ell(A_{S,r}(x_z), y_z).$$

We are interested in bounding the *generalization error*

$$\left| R(A_{S,r}) - R_e(A_{S,r}) \right|. \tag{1}$$

When talking about SGD we omit $A$ and just use $R(S, r)$ and $R_e(S, r)$ to represent $R(A_{S,r})$ and $R_e(A_{S,r})$ respectively.

**About the loss function** $l(\cdot, \cdot)$**.** When we talk about the loss function we refer to the commonly used loss functions in machine learning, like cross-entropy, focal loss, mean squared error (for bounded inputs) etc. Our results are valid for any bounded value loss function which is doubly differentiable. In machine learning an implicit assumption is that the algorithm is able to successfully minimize the loss function chosen, i.e., a loss function is used that can be minimized to a reasonable value over a training set. We also work with this assumption.

## 3.2 A Weakening of Uniform Stability

Given $S = \{Z_1, \ldots, Z_m\}$ where all points are chosen randomly from $D$, we construct $S^i$ via replacing the $i$-th element of $S$ by an independently generated element from $D$. To quote it formally we choose $\{Z_{1+m}, \ldots, Z_{2m}\}$ points such that all are chosen randomly from $D$ such that they are independent from all points in $S$. For each $i \in [m]$ we define

$$S^i = \{Z_1, \ldots, Z_{i-1}, Z_{i+m}, Z_{i+1}, \ldots, Z_m\},$$

where $[m]$ represents integer points from $[1, m]$.

**Definition 3.1** (Almost Sure (a.s.) Support Stability)**.** We say an algorithm $A_{S,r}$ has *almost sure (a.s.) support stability $\beta$ with respect to the loss function $\ell(\cdot, \cdot)$* if for $Z_1, \ldots, Z_{2m}$ chosen i.i.d. according to an unknown distribution $D$ defined over $\mathcal{Z}$,

$$\forall i \in [m] : \forall z \in \mathrm{supp}\,(D) : \mathrm{E}_r\left[\left|\ell(A_{S,r}(x_z), y_z) - \ell(A_{S^i,r}(x_z), y_z)\right|\right] \leq \beta$$

with probability 1 over the choice of points $Z_1, \ldots, Z_{2m}$ where $\forall i, Z_i \sim D$ or in other words $\{Z_1, \ldots, Z_{2m}\} \sim D^{2m}$.

We note that this notion weakens the notion of uniform stability introduced by (Bousquet & Elisseeff, 2002, Definition 6) by requiring the bound on the difference in losses to hold $D^{2m}$- almost everywhere. This probability is defined over the random choices of $Z_1, \ldots, Z_{2m}$. Besides the condition on the loss is required to hold only for those data points that lie in the support of $D$. These conditions make a.s. support stability a *data-dependent quantity* on the lines of the suggestion made by Zhang et al. (2017). We also observe that a.s. support stability is comparable to but stronger than the hypothesis stability of Kearns & Ron (1999) as formulated by Bousquet & Elisseeff (2002).

While the quantification of $z$, i.e., $\forall z \sim \mathrm{supp}\,(D)$ appears to be a very strong condition it is a weakening of uniform stability. In (Bousquet & Elisseeff, 2002, Section 5) it was shown that uniform stability (which is $\forall z \sim D$) holds for several classical machine learning algorithms like soft margin SVM, bounded SVM regression and regularized least square regression. Hence a.s. support stability also holds for these algorithms. As we will see ahead, the weakening helps us fulfil key technical requirements when it comes to the study of neural networks.

## 3.3 Exponential Convergence of Generalization Error

Almost Sure (a.s.) Support Stability can be used in place of uniform stability in conjunction with the techniques of (Feldman & Vondrak, 2019a, Theorem 1.1) to give guarantees on generalization error for algorithms that are *symmetric in distribution*. A function $f(x_1, \ldots, x_m)$ is called symmetric if $f(x_1, \ldots, x_m) = f(\sigma(x_1), \ldots, \sigma(x_m))$ for any permutation $\sigma$. But if we have a function $f$ which is not symmetric but the probability of choosing any permutation of a given set of elements is equal then we use the term *symmetric in distribution* to refer to such a function along with the distribution by which its inputs are picked. In (Bousquet & Elisseeff, 2002, Section 2.1) the term "symmetric algorithm" was used but it was potentially misleading since what they meant was *symmetric in distribution* in the sense that we have used it. Since SGD randomly permutes the training points it is clearly *symmetric in distribution*.

In particular, we can derive the following theorem.

**Theorem 3.2.** *Let $A_{S,r}$ be an algorithm that is symmetric in distribution and has a.s. stability $\beta$ with respect to the loss function $\ell(\cdot, \cdot)$ such that $0 \leq \ell(A_{S,r}(x_z), y_z) \leq 1$ for all $S \in \mathcal{Z}^m$, for all $r \in \mathcal{R}$ and for all*

$z = (x_z, y_z) \in \mathcal{Z}$. *Then, there exists a constant $c > 0$ independent of $m$ s.t. for any $m \geq 1$ and $\delta \in (0, 1)$, with probability $1 - \delta$,*

$$E_r\left[R(S, r) - R_e(S, r)\right] \leq c\left(\beta \log(m) \log\left(\frac{m}{\delta}\right) + \sqrt{\frac{\log(1/\delta)}{m}}\right).$$

The constant $c$ is independent of $m$ and, because our analysis is asymptotic in $m$, this is sufficient for us.

*Proof outline.* We give a high-level outline here. Our proof extends the proof of Feldman and Vondrak ((Feldman & Vondrak, 2019a, Theorem 1.1)) to accommodate the generalization of McDiarmid's Lemma A.2 from (Combes, 2015, Proposition 2). Feldman & Vondrak (2019b) used two steps to get a better generalization guarantee. The first step is range reduction, where the range of the loss function is reduced. For this, they define a new clipping function in Lemma 3.1 Feldman & Vondrak (2019a) which preserves uniform stability and hence it will also preserve a.s. support stability. They also use uniform stability in Lemma 3.2 Feldman & Vondrak (2019a) where they show the shifted and clipped function will still be stable which is done by applying McDiarmid's inequality to $\beta$ sensitive functions. Here use a modification of McDiarmid's inequality (Lemma A.2 given in Appendix A) to get bounds for a.s. support stability. The second step is dataset size reduction (as described in Section 3.3 Feldman & Vondrak (2019a)) which will remain the same for a.s. support stability as this only involves stating the result for a smaller dataset and the probability, and then taking a union bound. Therefore both steps of the argument given in Feldman & Vondrak (2019a) go through for a.s. support stability.

## 4 Almost Sure (a.s.) Support Stability of Stochastic Gradient Descent

In this section, we show how to bound the a.s. support stability of a model whose parameters are learned by performing Stochastic Gradient Descent (SGD) using a training set. We will see that the bounds on stability can be formulated in terms of the gradients encountered during training and the value of the gradient at the end of the training. Since these gradients are determined by both the training set and the random choices made by SGD, this section shows that stability can be understood not just by looking at the data distribution, but by going beyond that and looking more carefully at the training process.

This section is organized as follows:

**Section 4.1.** We define three quantities associated with SGD that will be used to bound a.s. support stability: Worst Case Training Gradient (WCTrG), the Test Gradient (TeG) and Second Moment of Step Training Gradient (SMSTrG). We will also show that these quantities are bounded under certain mild conditions in Section 4.1.2.

**Section 4.2.** We will show a bound on a.s. support stability that depends on WCTrG and TeG. This bound is weak in the sense that WCTrG is related to the largest of the gradients encountered during training and strong in the sense that the stability bound decreases as $m^{-1}$.

**Section 4.3.** We will show a bound on a.s. support stability that depends on SMSTrG and TeG for one epoch. This bound is stronger than the bound shown in Section 4.2 in the sense that SMSTrG is associated with an averaging of the gradients encountered during training rather than the largest of the gradient. However, the bound is weaker in the sense that it decreases as $m^{-1/2}$.

**Section 4.4.** This section is devoted to making the bound presented in Section 4.3 more useable by bounding Second Moment of Step Training Gradient in terms of the initial and final values of the loss function.

**Terminology.** We assume that the learned function is parameterized by a vector $\boldsymbol{w} \in \mathbb{R}^n$ for some $n \geq 1$, i.e., we have some fixed function $f : \mathbb{R}^n \times \mathcal{X} \to \mathcal{Y}$. The training set is used to learn a suitable parameter vector $\boldsymbol{w} \in \mathbb{R}^n$ such that the value $f(\boldsymbol{w}, x_z)$ is a good estimate of $y_z$ for all $z \in \mathcal{Z}$. We assume that this value of $\boldsymbol{w}$ is learned by running SGD using a training set drawn from the unknown distribution. We will say that

the size of the training set is $m$, and the algorithm proceeds in *epochs* of $m$ steps each. The parameter vector at step $t$ is denoted by $\boldsymbol{w}_t$ for $0 \le t \le \tau \cdot m$, where $\tau$ is the total number of epochs during training.

To frame the learned function output by this algorithm in the terms defined in Section 3.1, the random decision string $r$ consists of the pair $(\boldsymbol{w}_0, \boldsymbol{\pi} = (\pi_0, \ldots, \pi_{\tau-1}))$ which we also write as $(r_{init}, r_p)$, i.e., the random initial parameter vector $\boldsymbol{w}_0$ (or $r_{init}$) from which SGD begins and the sequence of $\tau$ random permutations used in the $\tau$ epochs. The weights encountered at step $t$ using permutation $\boldsymbol{\pi}$ will be represented as $\boldsymbol{w}_{t,\boldsymbol{\pi}}$. For the sake of brevity, except for Section 4.1.1 we omit the use of $\boldsymbol{\pi}$ and only write $\boldsymbol{w}_t$.

## 4.1 Some quantities associated with SGD gradients

### 4.1.1 Definitions

**Definition 4.1.** If $\Pi$ is the set of all permutations of $S$, then SGD as defined above has the following quantities associated with it

1. *Worst Case Training Gradient (WCTrG) $L_S$.*

$$L_{z,A} = \max_{\boldsymbol{w} \in A} \left\{ \left\| \frac{\partial}{\partial \boldsymbol{w}} f(\boldsymbol{w}, z) \right\| \right\}.$$

   Where the set $A = \cup_{\boldsymbol{\pi} \in \Pi^\tau} \{\boldsymbol{w}_{1,\boldsymbol{\pi}}, \ldots, \boldsymbol{w}_{\tau m-1, \boldsymbol{\pi}}\}$, i.e. all the parameter vectors encountered across all time steps across all possible permutations of $S$. $L_S = \max_{z \in S} \{L_{z,A}\}$ which is a random variable depending on the choice of $S$ and $\boldsymbol{w}_0$.

2. *Training Smoothness Constant $K_S$.*

$$K_{z,A} = \max_{\substack{\boldsymbol{w}, \boldsymbol{w}' \in A^2 \\ \boldsymbol{w} \neq \boldsymbol{w}'}} \left\{ \frac{\|f'(\boldsymbol{w}, z) - f'(\boldsymbol{w}', z)\|}{\|\boldsymbol{w} - \boldsymbol{w}'\|} \right\}.$$

   Where the set $A = \cup_{\boldsymbol{\pi} \in \Pi^\tau} \{\boldsymbol{w}_{1,\boldsymbol{\pi}}, \ldots, \boldsymbol{w}_{\tau m-1, \boldsymbol{\pi}}\}$. We will say that $K_S = \max_{z \in S}\{K_{z,A}\}$. Note that $K_S$ is a random variable depending on the choice of $S$ and $\boldsymbol{w}_0$.

3. *Test Gradient (TeG) $L_g$.*

$$L_{z,g} = \max_{\substack{\boldsymbol{w}, \boldsymbol{w}' \in A^2 \\ \boldsymbol{w} \neq \boldsymbol{w}'}} \left\{ \frac{\|f(\boldsymbol{w}, z) - f(\boldsymbol{w}', z)\|}{\|\boldsymbol{w} - \boldsymbol{w}'\|} \right\}.$$

   Where the set $A = \cup_{\boldsymbol{\pi} \in \Pi^\tau} \{\boldsymbol{w}_{\tau m, \boldsymbol{\pi}}\}$, i.e. all parameter vectors encountered at the end of the training process across all possible permutation of $S$. We will say that $L_g = \max_{z \in \mathcal{Z}}\{L_{z,g}\}$. Note that $L_g$ is a random variable depending on the choice of $S$ and $\boldsymbol{w}_0$.

4. *Second Moment of Step Training Gradient (SMSTrG) $\sigma_S$. This is defined for $\tau = 1$.*

$$\sigma_z = \sqrt{\frac{1}{m} \sum_{t=0}^{m} \left\| \max_{\boldsymbol{w} \in A_t} \left\{ \frac{\partial}{\partial \boldsymbol{w}} f(\boldsymbol{w}, z) \right\} \right\|^2}.$$

   Where the set $A_t = \cup_{\pi_0 \in \Pi} \{\boldsymbol{w}_{t,\pi_0}\}$ i.e. set of all possible parameter vectors encountered during training at step $t$ of SGD across all permutations. We say that $\sigma_S = \max_{z \in S}\{\sigma_z\}$. Note that $\sigma_S$ is a random variable in $S$ and $\boldsymbol{w}_0$.

Our stability analysis of SGD will depend on these quantities. However, when we take this analysis and try to apply it to NNs with discontinuous activation functions like ReLU, Training Smoothness Constant and Test Gradient defined above are not guaranteed to be bounded. So, it is necessary to establish the conditions under which they are bounded.

### 4.1.2 Boundedness

We now present the property under which the quantities defined above will be (almost surely) bounded.

**Definition 4.2** (Almost surely Locally Parameter Lipschitzness of parametrized functions)**.** A parameterized function $f : \mathbb{R}^n \times \mathcal{Z} \to \mathbb{R}$ is said to be *almost surely locally L-parameter Lipschitz* w.r.t $D$ if for a fixed $\boldsymbol{w} \in \mathbb{R}^n$ and for $z \sim D$ there exist constants $L > 0$ and $\epsilon > 0$ such that, with probability 1 (over the choice of $z$), for all $\boldsymbol{w}' \in \mathbb{R}^n$, $\|\boldsymbol{w}' - \boldsymbol{w}\| < \epsilon$ implies

$$\|f(\boldsymbol{w}', z) - f(\boldsymbol{w}, z)\| \leq L\|\boldsymbol{w}' - \boldsymbol{w}\|.$$

We will use the abbreviation *a.s. L-LPL* or simply *a.s. LPL* for a function that satisfies Definition 4.2. Since training is always run for a finite number of steps it is easy to observe that *if the loss function is a.s. LPL, then $L_S$ and $\sigma_S$ are bounded.*

If $\nabla_w f(\cdot, \cdot)$ satisfies Definition 4.2 we will call such a function *almost surely locally L-parameter Smooth* or *a.s. L-LPS* for short. We now see that these two properties imply the almost sure boundedness of the quantities defined in Section 4.1.1. The important insight is that if the function (or its gradient) is locally bounded, and, if we only look at this function at a finite number of points, we get a "global" property within this finite set of points. We state this as a lemma.

**Lemma 4.3.** *If $f : \mathbb{R}^n \times \mathcal{Z} \to \mathbb{R}$ is bounded and a.s. $L_l$-LPL, and $A$ is a finite subset of $\mathbb{R}^n$, then with probability 1 there is an $L > 0$ such that for every pair $\boldsymbol{w}, \boldsymbol{w}' \in A$, if $z \sim D$, then*

$$\|f(\boldsymbol{w}, z) - f(\boldsymbol{w}', z)\| \leq L\|\boldsymbol{w} - \boldsymbol{w}'\|,$$

The proof is in Appendix B.

**Corollary 4.4.** *If $f : \mathbb{R}^n \times \mathcal{Z} \to \mathbb{R}$ is a.s. L-LPL then $L_g$ is almost surely bounded. Further, if $f$ is a.s. $K$-LPS then $K_S$ is almost surely bounded.*

The corollary follows by observing that due to the finiteness of the training process, the parameters on which the slopes are computed in Definition 4.1 are all drawn from finite sets.

## 4.2 Almost Sure (a.s.) Support Stability of SGD with WC TG

We now work towards the a.s. support stability of SGD using WCTrG and TeG. First, we state a theorem that bounds the replace-one error of SGD up to a certain number of epochs. To make the theorem statement easier to read, we first separate out our assumptions.

S1. We are given a space $\mathcal{Z} = \mathcal{X} \times \mathcal{Y}$ and a probability distribution $D$ defined over it. We have a parameterized loss function $f : \mathbb{R}^n \times \mathcal{Z} \to \mathbb{R}$ that is a.s. $L_l$-LPL w.r.t $\text{supp}(D)$ and a.s. $K_l$-LPS w.r.t $S$.

S2. For a training set $S$ of size $m$ for each $i \in [m]$ chosen i.i.d. according to $D$ we run SGD on $f$ for $\tau$ epochs with random decision string $r$. Parallelly, we do the same for set $S^i$ with the same set of random decision string $r$. For $S^i$, the $i$-th data point $z_i$ of $S$ has been replaced by another data point $z_i'$ chosen from $D$ independent of all other random choices.

S3. At step $t$ of SGD let the learning rate $\alpha_t \leq \alpha_0/t^{(1-\rho(\tau,m))}, \rho(\tau,m) = \frac{\log\log m}{\log \tau + \log m}$, $\boldsymbol{w}_t$ and $\boldsymbol{w}_t'$ the parameter vectors obtained for the $t$-th step of SGD while training with set $S$ and $S^i$ respectively.

**Theorem 4.5.** *Given assumptions S1, S2 and S3, we have $L_S$ as WCTrG, $L_g$ as TeG and $K_S$ as Training Smoothness Constant. Let the random decision string $r = (\boldsymbol{w}_0, \pi_0)$, then with probability 1 over $z$ we have,*

$$E_r\left[f(\boldsymbol{w}_{\tau m}, z) - f(\boldsymbol{w}_{\tau m}', z)\right] \leq 2^\tau \alpha_0 \cdot E_r\left[\frac{L_S L_g \cdot U(\alpha_0, K_S, \rho(\tau,m))}{m^{\left(1 - \frac{\alpha_0 K_S}{\rho(\tau,m)}\right)}}\right], \tag{2}$$

*where $U(\alpha_0, K_S, \rho(\tau,m)) \leq 1 + \frac{1}{K_S\alpha_0}$, and as $\alpha_0 \to 0$, $U(\alpha_0, K_S, \rho(\tau,m)) \to 1 + \frac{m^{\rho(\tau,m)}}{\rho(\tau,m)}$.*

Here, the expectation is over $r$ for random variables $L_S, L_g$ and $K_S$. Remember, the L.H.S. is still a random variable in $S$ and a.s. support stability is almost surely over this $S$.

*Proof outline.* The proof follows the lines of the argument presented by (Hardt et al., 2016, Theorem 3.12) with the difference that we allow for a probabilistic relaxation of the smoothness conditions and more relaxed constraint on gradient bounds in line with our definition of a.s. stability. Also, note that we have to account for an expectation over the random string $r$ and that we have been able to extend the argument to multiple epochs which was not possible in (Kuzborskij & Lampert, 2018, Theorem 4). The complete proof of Theorem 4.5 is in Appendix B.

*Data-dependence with WCTrG and TeG.* A key feature of the bound presented in equation 2 is that the dependence on the data is expressed through the data-dependent quantities WCTrG ($L_S$) and TeG ($L_g$). The WCTrG depends on the gradients at training points and the replacement point $z_i'$, which is also picked from the data distribution, and TeG depends on the gradients of the trained network calculated at points from distribution. The advantage of splitting TeG and WCTrG instead of a global bound is that TeG could potentially be very small when the loss is converged as it's calculated at the end of training. And, WCTrG only depends on training points instead of the entire distribution.

**Corollary 4.6.** *Given assumptions S1, S2 and S3, and under the condition that $K_S$ is a constant, w.r.t. $m$, for all $r$, and $E_r\left[L_g L_S\right]$ is also constant w.r.t $m$, there is a constant $c \in (0,1)$ that depends on $\alpha_0$ and $K_S$ such that if the number of epochs $\tau$ is at most $c \log m$ epochs, the expectation of the generalization error of the algorithm taken over the random choices of the algorithm decreases as $\tilde{O}\left(m^{-\min(\epsilon, 1/2)}\right)$ (where tilde hides the logarithmic factors) with probability at least $1 - 1/m$ over the choice of the training set if $\frac{\alpha_0 K_S}{\rho(1,m)} + c \log 2 < 1$, where $\epsilon = 1 - c \log 2 - \frac{\alpha_0 K_S}{\rho(1,m)}$.*

*Proof.* Let us consider two cases. In the first case when $\epsilon > 1/2$ (i.e. we get the usual rate $\tilde{O}(m^{-1/2})$), this happens when $\alpha_0 < \frac{\rho(1,m)}{2K_S}$ and we choose a small enough $c$. One the other hand for case where $\epsilon < 1/2$ (i.e. rate of $\tilde{O}(m^{-\epsilon})$), which allows for a larger learning rate $\frac{\rho(1,m)}{2K_S} < \alpha_0 < \frac{\rho(1,m)}{K_S}$ (for small enough $c$). This clearly shows that a larger initial learning rate could be bad for generalization. It is easy to check from Theorem 4.5 that with the conditions given in the statement of Corollary 4.6 the learning algorithm has a.s. support stability $\beta$ where $\beta$ is $o(1/m^\epsilon)$ if $\frac{\alpha_0 K_S}{\rho(1,m)} + c \log m < 1$. We can therefore apply Theorem 3.2 with $\delta = 1/m$ to get the result. $\square$

## 4.3 Almost Sure (a.s.) Support Stability of SGD with SMSTrG

We present a more fine-grained analysis that provides an alternate bound of Theorem 4.5 by replacing the WCTrG (Worst Case Training Gradient), which was defined for all SGD steps, with the SMSTrG (Second Moment of Step Training Gradient). *The significance of this is that this expected value could be substantially smaller than the worst case value in many cases, especially when the training converges rapidly from a high value of the loss to a low value.* We present the bounds using SMSTrG below.

**Theorem 4.7.** *Given the assumption S1, S2 and S3 for $\tau = 1$, we have $\sigma_S$ as SMSTrG, $L_g$ as TeG and $K_S$ as Training Smoothness Constant. Let the random decision string $r = (\boldsymbol{w}_0, \pi_0)$, then with probability 1 over $z$ we have,*

$$E_r\left[f(\boldsymbol{w}_m, z) - f(\boldsymbol{w}_m', z)\right] \leq \alpha_0 \cdot E_{\boldsymbol{w}_0}\left[\frac{L_g \cdot \sigma_S \cdot \sqrt{U'(\alpha_0, K_S, m)}}{m^{\frac{1}{2} - \frac{\alpha_0 K_S(\log m - 1)}{\log \log m}}}\right]. \tag{3}$$

*Where $U'(\alpha_0, K_S, m) \leq 1 + \frac{1}{2\alpha_0 K_S}$ and as $\alpha_0 \to 0$, $U'(\alpha_0, K_S, m) \to 1 + \frac{\log(m)^2}{\log \log m}$.*

*Proof outline.* The proof follows the proof of Theorem 4.5, but instead of using the WCTrG for bounding the gradients encountered at the step where the training points differ, we use the exact gradient at that step, i.e. $f(\boldsymbol{w}_{\pi_0(i)}, \cdot)$ for every permutation $\pi_0 \in \Pi$. Then, the proof continues till $m$ steps. Finally, when taking the expectation over the permutations we apply Cauchy Schwarz to separate other random variables and obtain the SMSTrG ($\sigma_S$). The complete proof of Theorem 4.7 is in Appendix B.

Although the bound in Theorem 4.7 looks weaker than the bound in Theorem 4.5 because of $m^{\frac{1}{2}}$ term, the key part here is the use of SMSTrG instead of WCTrG. SMSTrG could potentially be a lot smaller because it's averaged over the training steps. This will especially benefit the case when training converges fast and gradients quickly become small. We later in Section 4.4 show that we can bound SMSTrG by the decrease in loss during training under some very mild conditions of assumption P1.

We now present the alternate version of Corollary 4.6 using Theorem 4.7.

**Corollary 4.8.** *Given assumption S1, S2 and S3 for $\tau = 1$, and under the condition that $K_S$ is constant w.r.t. $m$ for all $r = (\boldsymbol{w}_0, \pi_0)$, and $E_{\boldsymbol{w}_0} [L_g \cdot \sigma_S]$ is also constant w.r.t. $m$, the expectation of generalization error of algorithm taken over the random choices of the algorithm decreases as $\tilde{O}(m^\epsilon)$ (were tilde hides the logarithmic factors) with probability at least $1 - 1/m$ over the choice of training set if $\frac{\alpha_0 K_S (\log m - 1)}{\log \log m} < 0.5$, where $\epsilon = 0.5 - \frac{\alpha_0 K_S (\log m - 1)}{\log \log m}$.*

*Proof.* It's direct to see when we apply the conditions stated in the above corollary in Theorem 4.7 we get the desired result. □

## 4.4 Bound on SMSTrG

We now proceed to show the bound on SMSTrG (i.e. Second Moment of Step Training Gradient) along with the assumption required for the bound to hold. For this, we use a result from Bottou et al. (2018) and modify it slightly to show a bound on the expectation of SMSTrG over $S$ and $\boldsymbol{w}_0$. Although this is a bound on expectation, we later show in the discussion of Section 5.2.2 that we can use this as a high probability bound. And this is sufficient because of the probabilistic form of a.s. support stability.

First we define $F_S(\boldsymbol{w}_t) = \frac{1}{m} \sum_{i=1}^m f(\boldsymbol{w}_t, z_i)$. Now, in order to bound the expectation of SMSTrG over the training set, we use the result from (Bottou et al., 2018, Theorem 4.10) and modify it to get the required bound. This theorem, combined with Assumption P1, provides a bound on the expectation of SMSTrG over the training set (Corollary 4.10). The bound is in terms of the initial and optimal (final) values of the loss function (and includes terms based on the learning rate), so unless the initial loss is very bad, the bound will be useful. The theorem requires the following condition which is a kind of version of a bound on the variance of the gradients encountered during training.

**P1. (Assumption 4.3 (equation 4.9) of Bottou et al. (2018)).** There exist constants $M \geq 0$ and $M_G \geq 0$ (independent of the size of the training set) such that,

$$\frac{1}{m} \sum_{z \in S} \|\nabla f(\boldsymbol{w}_t, z)\|^2 \leq M + M_G \|\nabla F_S(\boldsymbol{w}_t)\|^2. \tag{4}$$

**Theorem 4.9.** *(Bottou et al., 2018, Theorem 4.10) Suppose we run SGD under the assumption that $K_S$ is constant for all $r$ and P1 holds, for $M \geq 0$, $M_G \geq 0$. If $\boldsymbol{w}_0$ is the initialization weight, $\boldsymbol{w}^*$ is optimal value for $F_S(\boldsymbol{w})$ and $\alpha_t$ is any diminishing step size then,*

$$\sum_{t=1}^T \alpha_t E \left[ \|\nabla F_S(\boldsymbol{w}_t)\|^2 \right] \leq 2(E[F_S(\boldsymbol{w}_0)] - E[F_S(\boldsymbol{w}^*)]) + E[K_S] M \sum_{t=1}^T \alpha_t^2, \tag{5}$$

*where $F_S(\boldsymbol{w}_t) = \frac{1}{m} \sum_{i=1}^m f(\boldsymbol{w}_t, z_i)$.*

Now, using Theorem 4.9 and assumption P1 we can get a bound on the expectation of SMSTrG. We state this in the next corollary.

**Corollary 4.10.** *If we run SGD for $1$ epoch under the assumption that $K_S$ is constant for all $r$ and P1 holds, for $M \geq 0$, $M_G \geq 0$, $\boldsymbol{w}_0$ is the initialization weight, $\boldsymbol{w}^*$ is optimal value for $E_{z \in S} [f(\boldsymbol{w}, z)]$, $r = (\boldsymbol{w}_0, \pi_0)$ is the random string for SGD and $\alpha_t$ is any diminishing learning rate then,*

$$E_{S,\boldsymbol{w}_0} [\sigma_S] \leq M + 2M_G E[f(\boldsymbol{w}_0, z) - f(\boldsymbol{w}^*, z)] + \frac{E[K_S] M_G M}{\alpha_0} \sum_{t=0}^T \alpha_t^2 \tag{6}$$

The proof of this theorem is in Appendix B at the end. The R.H.S of this equation is constant w.r.t. $m$ if $\sum_{t=1}^{T} \alpha_t^2$ is constant w.r.t. $m$ and in our analysis this is true (c.f. assumption S3). Hence, this corollary becomes useful for us since it shows that we can bound the expectation of SMSTrG under a mild assumption.

# 5 Neural Networks with ReLU Activation

We now proceed to show what the a.s. support stability bounds on SGD translates into for neural networks. This includes two generalization bounds, one using WCTrG (i.e. Worst Case Training Gradient) in Section 5.1 and the other using SMSTrG (i.e. Second Moment of Step Training Gradient) in Section 5.2. We also emphasise on the conditions required for generalization bounds to hold and translate those conditions to more practical constraints.

## 5.1 Almost Sure (a.s.) Support Stability of Neural Networks with ReLU Activation: WCTrG

First, we present the generalization bound of neural networks via a.s. support stability using WCTrG (Worst Case Training Gradient) in Section 5.1.1. We then, in Section 5.1.2, show the conditions required for the bounds to hold. We also present a discussion at the end highlighting an example where the actual data dimension is small, and our gradient bounds will be better.

### 5.1.1 Generalization Using WCTrG.

For ease of reading, we first state our assumptions and then, in Theorem 5.1 we present the bound.

**N1.** We have a fully connected neural network with ReLU activation and 1 output neuron.

**N2.** The NN is trained on set $S \sim D^m$ using SGD for $\tau$ epochs, where $D$ is over $\mathbb{R}^d \times \mathcal{Y}$, such that $\mathcal{Y}$ is countable and for each $y \in \mathcal{Y}$ we get a countable set $\{x \in \mathbb{R}^d : \Pr_D \{\mathsf{lab}(x) = y\} > 0\}$, where $\mathsf{lab}(x)$ is label of x.

**N3.** We have a doubly differentiable loss function with bounded first and second order derivatives and learning rate $\alpha_t = \alpha_0/t^{(1-\rho(\tau,m))}$, where $\rho(\tau, m) = \frac{\log\log m}{\log \tau + \log m}$ and the data points of $S$ and the spectral norms of weight matrices explored by SGD are bounded

**Theorem 5.1.** *If N1, N2 and N3 hold, then there is a constant $c > 0$ such that*

$$E_r\left[\|R(S,r) - R_e(S,r)\|\right] \leq c\left(2^\tau \alpha_0 \cdot E_r\left[L_S L_g\right] \cdot U(\alpha_0, K_S, \rho(\tau, m))\frac{\log(m)^2}{m^{\left(1-\frac{\alpha_0 K_S}{\rho(\tau,m)}\right)}} + \sqrt{\frac{\log(m)}{m}}\right),$$

*with probability at least $1 - 1/m$, where $U(\alpha_0, K_S, \rho(\tau, m)) \leq 1 + \frac{1}{\alpha_0 K_S}$ and $\alpha_0 \to 0$ implies $U(\alpha_0, K_S, \rho(\tau, m)) \to 1 + \frac{m^{\rho(\tau,m)}}{\rho(\tau,m)}$.*

The proof of the above theorem is at the end of Section 5.1.2 before discussion. Note that for some $c_1 \log(m)$ epochs and with an initial learning rate of $\alpha_0$ such that $\frac{\alpha_0 K_S}{\rho(1,m)} + c_1 \log 2 < 1$, the RHS decreases as $m$ increases. It is important to note that TeG ($L_g$) is the constant that depends on the actual distribution $D$ and is calculated for a trained neural network (i.e. at $\boldsymbol{w}_{\tau m}$). This aligns with the notion that if the network has reached a "good enough" minima, then the gradient values should be less; hence, this will show better generalization. Also, WCTrG ($L_S$) and Training Smoothness Constant ($K_S$) depend on the training set $S$. These are "global" over the data set in the sense that the expectation is for the entire training process over the (random) choice of initial parameters and the permutation that SGD chooses. For cases where SGD chooses a good set of initial parameters with good probability, these are likely to be small.

For Training Smoothness Constant ($K_S$), we are constrained in the sense that we need this value to be small throughout the training, even at the beginning. Also it is interesting to note that when $L_S \to 0$ and $L_g \to 0$ the generalization error becomes zero, but when $K_S \to 0$, we have $U(\alpha_0, K_S, \rho(\tau, m)) \to 1 + \frac{m^{\rho(\tau,m)}}{\rho(\tau,m)}$ which

leads to generalization error behave like $O\left(\frac{\log m^3}{m} + \sqrt{\frac{\log m}{m}}\right)$. Although this is still a decreasing function of $m$, this does not directly go to zero. This highlights that unlike $L_S$ and $L_g$, just $K_S \to 0$ alone is insufficient for zero generalization error.

The role of gradients in bounding generalization error that we identify has the same flavour, in a very different context, as the work of (Negrea et al., 2020b, Theorem 3.1) and (Haghifam et al., 2020, Theorem 4.2) where Information-Theoretic generalization bounds for SGLD (Stochastic Gradient Langevin Dynamics) are given in terms of gradient values of the loss function. Next in Section 5.1.2, we first establish that the theory of a.s. support stability applies to NNs under conditions specified, then we prove the Theorem 5.1 along with empirical validation.

### 5.1.2 Conditions on NNs for Generalization

The key to showing the a.s. support stability of NNs with ReLU is to establish that they are locally parameter-Lipschitz and locally parameter-smooth. First, we show the existence of these constants. Then we will show an upper bound under some reasonable assumptions and finally present an example in discussion.

**Theorem 5.2.** *For every $\boldsymbol{w} \in \mathbb{R}^n$, a doubly differentiable loss function, $\ell : \mathbb{R} \times \mathbb{R} \to \mathbb{R}$, applied to the output of a NN with ReLU activation is locally parameter-Lipschitz and locally parameter-smooth for all $x \in \mathbb{R}^d$ except for a set of measure 0.*

*Proof outline.* The proof of this theorem is based on the argument that for a given $\boldsymbol{w}$ a point of discontinuity exists at a given neuron if the input $x$ lies in the set of solutions to a family of equations, i.e., in a lower dimensional subspace of $\mathbb{R}^d$. This proof is an adaption of an idea of (Milne, 2019, Lemma 1) and can be found in Appendix C.2.

Theorem 5.2 begs the question: How large are these WCTrG, TeG and Training Smoothness Constant? We provide some general bounds that can be improved for specific architectures:

**Proposition 5.3.** *Suppose we have a fully connected NN of depth $H + 1$, with ReLU activation at the inner nodes. Then, if the spectral norms of weight matrices are bounded for every layer i.e., $\|W^i\|_\sigma$ is bounded $\forall i \in [H]$, and the size of each layer be $\{l_0, \ldots, l_H\}$ and the distribution of dataset is normalized with $\|x\|_2 \leq 1$ then,*

$$L_g \leq \left(\prod_{k=1}^{H} \|W^k\|_\sigma\right) \times \mathcal{A}(M, W)^{1/2} \tag{7}$$

$$K_S \leq \left(\prod_{k=1}^{H} \|W^k\|_\sigma\right) \times \mathcal{A}(M, W) \tag{8}$$

*where*

$$\mathcal{A}(M, W) = \sum_{l=1}^{H} \frac{\|M^l\|_{2,2}^2}{\|W^{l-1}\|_\sigma^2 \cdot \|W^l\|_\sigma^2 \cdot \|W^{l+1}\|_\sigma^2}$$

*where $(i, j)^{th}$ element of matrix $M^l[i, j] = \|M'(l, i, j)\|_\sigma$, and where $M'(l, i, j)$ is a matrix such that $(p, q)^{th}$ element is $M'(l, i, j)[p, q] = w_{j,p}^{(l+1)} w_{q,i}^{(l-1)}$. Note that equation 7 holds for both WCTrG ($L_S$) and TeG ($L_g$).*

The proof of the proposition is in Appendix C.2. Note that it's possible to give a tighter bound for the above theorem by not bounding the product of weight matrices (which we do after equation 19 in Appendix), but we keep the above equation because of its clarity. The bounds on these should be compared to the bounds given in the context of Rademacher complexity by (Bartlett et al., 2017, Equation 1.2) and Golowich et al. (2018). Our bound is related to the spectral complexity and can potentially be independent of the size of the network. We are now ready to prove our main theorem.

*Proof of Theorem 5.1.* Theorem 5.2 tells us that a NN with ReLU activations is locally parameter-Lipschitz and locally parameter-smooth. From Proposition 5.3, we see that the boundedness of the first and second derivatives of the loss function and the boundedness of the spectral norm of weight matrices and data points ensures that the gradient bound and smoothness constants associated with the NN's training are bounded w.r.t. $m$. With all these in place, we can apply Theorem 4.5 to get the a.s. support stability followed by Theorem 3.2 to get the desired result. $\qquad\square$

**Discussion: An example showing the benefits of data-dependent gradient bounds.** In general, data-dependent gradients (WCTrG and TeG) can be much smaller than the global Lipschitz constant of the space from which the data might appear. There are probably many scenarios in which this can be demonstrated, but we turn to a well-appreciated scenario: a data set that has much smaller dimensionality than the space in which it is embedded. We will now argue that in such scenarios, data-dependent gradients (WCTrG and TeG) can be significantly smaller.

Suppose we have data as $x \in \mathbb{R}^d$, but the actual dimension of the data is $D \ll d$, a situation that is often seen in many cases, for example, image data. For simplicity of presentations, we assume that each data point has $x_1, \ldots, x_D$ non-zero and the remaining coordinates are 0. The arguments we make can be made even without this assumption by considering the data points with coordinates based on their projection onto a basis of the subspace they are taken from.

Suppose we have a neural network with 1 hidden layer of $d_1$ neurons and a single output layer. Let $W^1 \in \mathbb{R}^{d_1 \times d}$ and $W^2 \in \mathbb{R}^{1 \times d_1}$ be the weights of $1^{st}$ and $2^{nd}$ layer respectively and we use $w_{i,j}^{(l)}$ to represent $i, j$ weight of $l^{th}$ layer. For simplicity, let the output of the neural network $O(\boldsymbol{w}, x) = W^2 W^1 x$. Now, assuming MSE loss, we calculate the gradients and show that the effective upper bound of this could be smaller because of the fact that our WCTrG and TeG are calculated only from $S$ and $supp(D)$. This is under the assumption that the weights are upper bounded by some quantity $B_1$. We will also assume that all data points have been re-scaled so that their norm is at most 1.

**Theorem 5.4.** *If the above conditions holds then we have a bound on $\ell_2$ norm of the gradients of the parameter vector $\boldsymbol{w}$ as,*

$$\|\nabla_{\boldsymbol{w}} f(\boldsymbol{w}, x)\|_2^2 \le B_1^2 \cdot d_1 \cdot D(1 + D)$$

Proof of the Theorem 5.4 is in Appendix C.2.

Here, we see that we obtain a bound on the norm of the gradients that are related to $D$, which is significantly smaller than $d$, whereas in general, we can expect the norm of the gradients to be of the order of $d$ even under the assumptions of bounded weights and rescaled data points.

Note that we show here the value of WCTrG ($L_S$) and TeG ($L_g$), but for generalization, we actually need $\mathbb{E}_r[L_S \cdot L_g]$ to be bounded. Using Cauchy-Schwarz inequality we could see that we need bound on just the expectation of square (or the second moment) of each term. This means that when we select the initial weight parameter vector $\boldsymbol{w}_0$, we need the boundedness constraints on weights only, which is a fairly mild constraint.

## 5.2 Almost Sure (a.s.) Support Stability of Neural Networks with ReLU Activation: SMSTrG

This Section focuses on stating the generalization bound of neural networks via a.s. support stability using the SMSTrG (i.e. Second Moment of Step Training Gradient). It's already shown in Bottou et al. (2018) that even for non-convex cases, the gradients of SGD decrease as the training proceeds. But earlier analysis and even Section 5.1 could not use this fact. So, we use the SMSTrG (i.e. Second Moment of Step Training Gradient), which will help us exploit this fact and provide a generalization bound for NNs. In Section 5.2.2, we present the proof and discuss the conditions under which our bound holds. We also present a real-world example highlighting the applicability of our results.

### 5.2.1 Generalization Using SMSTrG

We use the same conditions as defined in Section 5.1.1. Now, applying this to NNs and using SMSTrG, we get a generalization bound, which is an alternate version of Theorem 5.1 for the 1-epoch case.

**Theorem 5.5.** *If N1, N2 and N3 hold there is a constant $c > 0$ such that for all $i \in [m]$*

$$E_r\left[\|R(S,r) - R_e(S,r)\|\right] \leq c \left( \alpha_0 \cdot \sqrt{E_{\boldsymbol{w}_0}\left[L_g^2\right] E_{\boldsymbol{w}_0}\left[\sigma_S\right] \cdot U'(\alpha_0, K_S, S)} \frac{\log(m)^2}{m^{\left(\frac{1}{2} - \frac{\alpha_0 K_S(\log m - 1)}{\log\log m}\right)}} + \sqrt{\frac{\log(m)}{m}} \right),$$

*with probability at least $1 - 1/m$, where $U'(\alpha_0, K_S, m) \leq 1 + \frac{1}{2\alpha_0 K_S}$ and as $\alpha_0 \to 0$, $U'(\alpha_0, K_S, m) \to 1 + \frac{\log(m)^2}{\log\log m}$*

Note the presence of SMSTrG ($\sigma_S$) on the R.H.S here as opposed to WCTrG ($L_S$) in the statement of Theorem 5.1. We propose this is a much more practical quantity to bound. We have already shown in Section 4.4 an expectation bound on SMSTrG by the difference between the initial and optimal values of the loss function.

### 5.2.2 Conditions on NNs to Hold for Generalization

*Proof outline of Theorem 5.5.* Using Corollary 4.10 we already have a bound on expectation of SMSTrG. So instead of Proposition 5.3, we use this Corollary to achieve the result. Although note that the result is in expectation over the training set, we can simply apply Markov inequality as done in the discussion after Theorem 5.6 to get a probability bound. This is sufficient because of the probabilistic form of a.s. support stability. Also, the bound on the expectation of SMSTrG required only assumption (P1), which is a relatively mild assumption.

It's important to note that, like in the previous case (Section 5.1.2), we required the spectral norm of weight matrices to be bounded, but it's a difficult condition to ensure while training a neural network. On the other hand, the bound on the expectation of SMSTrG depends on the difference between the initial and optimal loss values, which is very easy to ensure and verify. This means we get a reasonable bound unless we have a very bad initial loss. Moreover, the assumption P1 in Section 4.4 is also intuitive as a very bad training point could badly affect the training, so bounding the variance of gradients (that too with respect to the average of gradients) restricts such bad points to be present in the distribution.

We now move on to show the applicability of our bound for SMSTrG.

**Discussion: Applicability in Classification with Two Layer Neural Network.** We now use Corollary 4.10 to establish a generalization bound for the case of two-class classification with a two-layer Neural Network. For simplicity of exposition, we have assumed that the data points are taken from $\mathbb{R}^2$. We fully specify the problem through the following assumptions:

**X1.** We have a two class classification ($\mathcal{Y} = \{-1, 1\}$) in 2 dimension ($x = [x_1, x_2]$) such that for expectations of centers we have $\mathrm{E}[x_1|y=1] = \mathrm{E}[x_2|y=1] = 2$ and $\mathrm{E}[x_1|y=-1] = \mathrm{E}[x_2|y=-1] = -1$, for second moment for a constant $\sigma > 0$ we have $\mathrm{E}[x_1^2] = \mathrm{E}[x_2^2] = \sigma^2$ and also for a constant $\mu_p > 0$ $\mathrm{E}[|x_1|] = \mathrm{E}[|x_2|] = \mu_p$.

**X2.** We use a single hidden layer feed-forward neural network, with $k$ as hidden layer size. Its parameters are initialized from $w_{i,j}^{(l)} = \mathcal{N}(0, 1)$. The total number of parameter values are $n = 2k + k$. For a loss function $f(\boldsymbol{w}, z) = |y - O(\boldsymbol{w}, x)|$, where $O(\boldsymbol{w}, x)$ is the output of the neural network, $\nabla f(\boldsymbol{w}, z)_j$ denotes the $j$-th partial derivative of function $f$ and $w_{j_2, j_3}^{j_1}$ be the associated weight. Let $\alpha_t$ be any diminishing step size at step $t$ of SGD.

**X3.** We assume the ratio of absolute values of the weights of the Neural Network are bounded by $B$ where $B > 1$.

**Theorem 5.6.** *If X1,X2,X3 hold, for $r = (\boldsymbol{w}_0, \pi_0)$ we have,*

$$E_{S,\boldsymbol{w}_0}[\sigma_S] \leq \frac{16B^4\sigma^2\left(\pi + 4k\sigma^2\mu_p\right)}{(B-1)^2\pi} \tag{9}$$

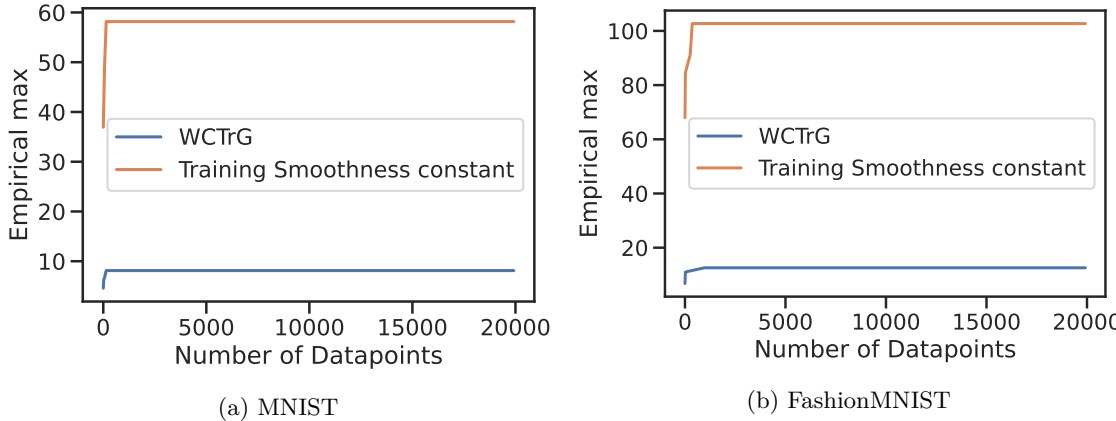

(a) MNIST  (b) FashionMNIST

Figure 1: Experiment 1, Maximum of the Worst Case Training Gradient (WCTrG) and Training Smoothness Constant at every $p$ $(= 20)$ interval of updates of SGD (we plot both the running average and the highest value found so far). Notice that these constants have a clear upper bound throughout the training process.

The proof of this theorem is in Appendix C.3. Now we can directly use Markov inequality to get a high probability bound. Assuming R.H.S of above is $B'$, so we will have,

$$\Pr_S \left\{ \mathbb{E}_{\boldsymbol{w}_0} \left[ \sigma_S \right] \geq \frac{16 B^4 \sigma^2 \log m \times (k\pi + 4k^2\sigma^2\mu_p)}{(B-1)^2\pi} \right\} \leq \frac{1}{k \log m}$$

Since the distance between the centers of the two classes is fixed at 2, the bound presented here satisfies our intuition by showing that if the variances of the two classes are small, i.e., the classes are well-separated, then the expectation of SMSTrG is small, and hence the generalization bound of Theorem 5.5 is small. It's important to note that since we have a probability bound, we need to apply the probabilistic version of support stability, although we show generalization only through a.s. support stability. The probability version can easily be derived using Theorem A.2 as we discuss in Section 3.3.

# 6 Experiments and Empirical Results

We now proceed to verify the results in a real-world case empirically. First, we perform the experiment to show that WCTrG, SMSTrG and TeG are bounded. We also show in the case of random labelling that our TeG are not bounded throughout the training, and hence, our results do not imply good generalization as expected.

## 6.1 Experimental Validation of Results

Here, we will experimentally show that the WCTrG ($L_S$), TeG ($L_g$) and Training Smoothness Constant ($K_S$) that we reasoned with are indeed bounded and that the theoretical upper bound that we derived for the generalization error of a neural network holds in practice. Note that if $L_S$ is bounded, then $\sigma_S$ will also be bounded. For simplicity in this experiment, we assume WCTrG to be a good proxy for TeG (as in general $L_g \leq L_S$).

**Setup.** For our experiments we use *MNIST* and *FashionMNIST* datasets. In both datasets, we randomly selected $20,000$ training and $1,000$ test points. All experiments were conducted using a fully connected feed forward neural network with a single hidden layer and ReLU activation. We train the model using SGD (batch size = 1), with cross-entropy loss, starting with randomly initialized weights. As suggested in our analysis we use a decreasing learning rate $\alpha_t = \frac{\alpha_0}{t}$. In each epoch, we consider a random permutation of the training set. WCTrG and Training Smoothness Constant are computed by calculating the norm of gradients and Hessian across the training steps and taking their max.

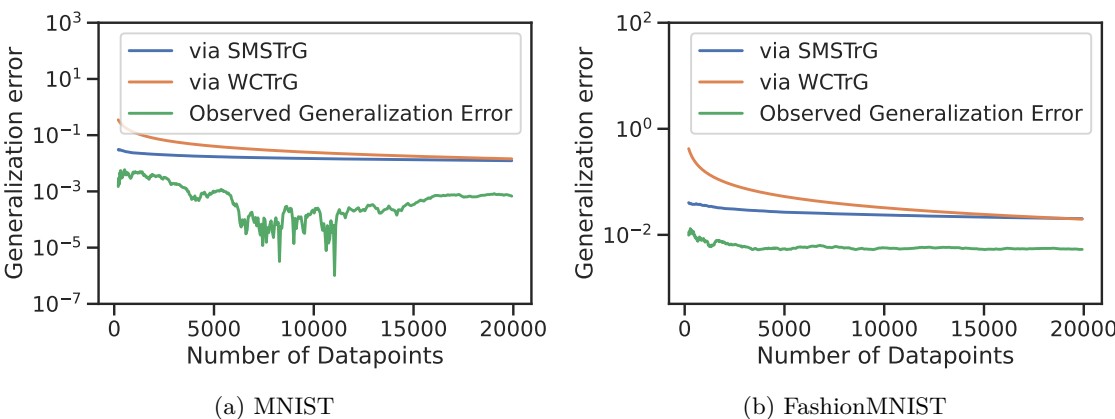

(a) MNIST

(b) FashionMNIST

Figure 2: Experiment 2, comparison of empirical generalization error (in green) vs. theoretical upper bound via WCTrG (in red) and via SMSTrG (in blue) with varying training set size for different datasets.

**Experiment 1.** Our first experiment is aimed towards establishing that the WCTrG ($L_S$) and Training Smoothness Constant ($K_S$) values estimated using local values at each step are bounded. Figure 1 summarizes the results of these experiments over MNIST and FashionMNIST datasets ($\alpha_0 = 0.001$). The plots contain the maximum of the gradients and smoothness values obtained after running each experiment 10 times with random weight initialization. These results support our Theorem 5.2 since the upper bound values quickly stabilize and do not grow with the size of the training set in both datasets. Similarly, the bounded smoothness constant supports our constraint on the learning rate, $\alpha_0 \leq \frac{\rho(\tau,m)}{K_S}, \rho(\tau, m) = \frac{\log \log m}{\log \tau + \log m}$. We find $L_S$ to be 8.1174 (MNIST) & 12.5737 (FashionMNIST), and $K_S$ to be 58.185 (MNIST) and 102.7096 (FashionMNIST).

**Experiment 2.** We now turn our attention to the experiment to support our main result, i.e., the empirical generalization error estimated using the validation set is upper bounded by our theoretical upper bound. We first split each dataset in a 20:1 ratio into training and validation sets and train the model at varying sizes of training sets. We empirically compute the generalization error at each training set size using the validation set. Figure 2 compares this empirical generalization error (in green) vs. the theoretical upper bound using WCTrG (in red) and using SMSTrG (in blue). From these results, we can see that our bound decreases along with the generalization error, thus empirically validating our reasoning. But clearly, the bound is not as tight as we would like it to be. We conjecture that this is because of the upper bounding of the $U(\cdot)$ term and the maximum which is taken across permutations of weights even for SMSTrG at $t$-th step.

## 6.2 Random Labelling Case

In making their case against the applicability of uniform stability as a tool for theoretically establishing the good generalization properties of neural networks, Zhang et al. (2017) presented the following classification problem: Given points picked from Euclidean space using some well-behaved distribution, say a Gaussian, each point was assumed to have a class label picked uniformly at random from a finite set of labels independent of all other points. Clearly, any classification algorithm trained on a finite training set will have $\omega(1)$ generalization error for this problem. We now demonstrate that our results *do not* imply good generalization for this problem. Specifically, we show empirically that the assumption of TeG ($L_g$) being independent of $m$ breaks in this case and this "constant" actually increases with $m$.

**Setup.** We pick images from the 0 and 1 label class of MNIST dataset. For random labelling case, we assign random labels to all the points. We then randomly sample a test set $\mathcal{T}$ ($|\mathcal{T}| = 50$). We take a single hidden layer (128 neurons) fully connected neural network having ReLU activation in the hidden layer. We take the loss function as $l(\hat{y}, y) = 1 - \text{Softmax}(c \cdot \hat{y}, y)$ where $c = 6$. We use a constant learning rate of 0.003, batch size of size 8.

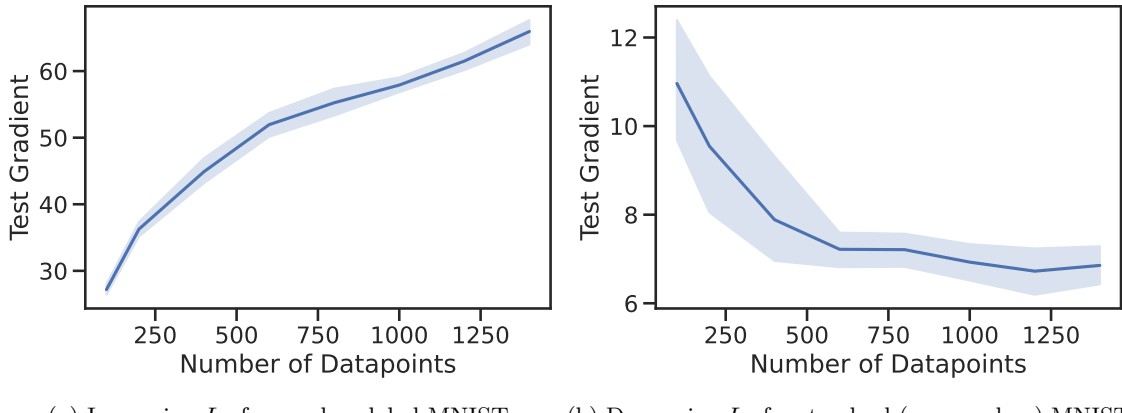

(a) Increasing $L_g$ for random label MNIST    (b) Decreasing $L_g$ for standard (non random) MNIST

Figure 3: Experiment 3, Test Gradient ($L_g$) plot as training set size increases.

**Experiment 3.** The experiment proceeds by selecting initial random weights for a model say $\boldsymbol{w}_0$ (we do this 10 times). Then for every initialization we pick training set $S$ from our modified dataset (we do this for 5 times). Now for every training set, we train the model either till accuracy is $\geq 98\%$ or till 500 epochs whichever is reached first. Now we calculate the loss i.e. $f(r, S, z)$ and the gradient $\nabla_{\boldsymbol{w}} f(r, S, z)$ for all $z \in \mathcal{T}$. For the TeG we do $L_g \simeq \max_{z \in \mathcal{T}}\{\|\nabla_{\boldsymbol{w}} f(r, S, z)\|\}$. In figure 3a we can clearly see that the TeG ($L_g$) scales as the size of the training set ($m$) increases. On the contrary for the standard (non-random) dataset the TeG shows a decreasing trend with $m$ see figure 3b. Therefore we can expect that in the random labelling case, the upper bound in Theorem 5.1 becomes so large as to become vacuous.

**Discussion.** We note that the random labelling example has the property that the variance over the choice of training sets of the loss of any algorithm, $Var_S[f(r, S, z)]$, is bound to be high. One possible direction for theoretically showing that this implies that the TeG are likely to be high is by using Poincare-type inequalities. This shows that the norm of the gradients of a function of a random vector is lower bounded by the variance of the function. We do not pursue this direction further here, but we point out that it may help develop a general theory for the limitations of what can be learned using parametrized methods trained using gradient descent methods.

## 7    Applicability and Conclusion

### 7.1    Discussion on the Applicability of Our Results

• *Removing the fully connectedness constraint.* Although we considered a fully connected network for Theorem 5.2 and Proposition 5.3, our data-dependent quantities are independent of the architecture of Neural Networks. We only provide a bound on WCTrG and TeG using this. Note that even the bound on SMSTrG is independent of the architecture of the network. We conjecture that it can be applied to architectures like CNNs which have partially connected convolution layers with intermediate pooling, normalization and skip connections (e.g., ResNet, DenseNet, etc). Our work provides a framework in which the study of the gradients obtained during training such networks can help guide our understanding of their generalization properties.

• *Adding regularization terms to the loss function.* Several popular regularizers, the $\ell_2$ regularizer being a prominent example, are doubly differentiable and therefore Theorem 5.1 can be applied when such regularizers are used along with a doubly differentiable loss function. Here as well a mild addition for bound on derivative of regularization term in Theorem 5.3 may be able to help us prove results for this setting. However, it requires further investigation to establish such a result.

• *Activation functions apart from ReLU.* We present a comprehensive treatment of ReLU activation but we conjecture that results are not restricted to this kind of activation. Non-linearities like max-pool can also be

handled in our framework by proving that, like with ReLU, the points of discontinuity of such a non-linearity also lie in a set of Lebesgue measure 0. This provides a direction for future research in this area.

- *The case of multiple outputs* Although we state the Theorem 5.1 and Theorem 5.5 for the case of a NN with a single output, it is not difficult to extend the technique to cover the case of multiple outputs. However, this requires a full treatment, which we postpone to future work.

*What about other distributions?* The data-dependent gradient bounds (SMSTrG, WCTrG and TeG) turn out to be the deciding factor of generalization error. However, our analysis is limited to the bounds we derive for them. There is a requirement for a more fine-grained analysis of these gradient bounds and we believe that optimizing these data-dependent gradient bounds will be the right direction to proceed. This may be made possible by looking at the network structures, the data distribution and the training set in more detail. We hope that the polynomial characterization of the NN presented in Appendix C.1.2 will help this process. We conjecture that it may be able to show that for certain distributions, the constants actually improve (decrease) w.r.t. training set size as the training proceeds, resulting in a much slower decay of learning rate and this could lead to a proof of a.s. support stability in these cases.

## 7.2 Conclusion

Using a.s. support stability, we derive the generalization error bounds for neural networks. We most importantly identify the data-dependent quantities (WCTrG, SMSTrG, TeG) whose boundedness implies generalization of neural networks. We also show how to upper bound these quantities either in terms of *spectral property* or *the initial loss and variance of gradients* of the neural network. So, this paper links the generalization of neural networks directly with the gradients and shows a guarantee of better generalization if we start with a small (constant w.r.t. training size) initial value of loss function and descent (defined in assumption P1) value of variance of the gradient at each step of SGD. However, we feel it is possible to prove stronger and more widely applicable results in this framework than the ones we have presented here. Immediate lines of research are to apply our methods for CNNs and GNNs and to investigate what other architectures can be approached with our method and whether the gradient bounds play some significant role because of a different network structure. Also, we do provide empirical evidence to support the example of the random labelling case but it lacks a theoretically concrete statement.

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

## A  Modification of McDiarmid's Theorem

We first define a probabilistic weakening of bounded difference property.

**Definition A.1.** Given $2m$ i.i.d. random variables $X_1, \ldots, X_{2m}$ drawn from some domain $\mathcal{Z}$ according to some probability distribution $D$, for some $\beta > 0$ and $\eta \in [0, 1]$, a function $f : \mathcal{Z}^m \to \mathbb{R}$ is called *$\eta$-almost $\beta$-bounded difference w.r.t. $D$* if

$$\forall i \in \{1, \cdots, m\} : |f(X_1, \ldots, X_m) - f(X_1, \ldots, X_{i-1}, X_i', X_{i+1}, \ldots, X_m)| \leq \beta,$$

with probability at least $1 - \eta$. In case $\eta = 0$ we say that $f$ satisfies *almost surely $\beta$-bounded difference w.r.t $D$*. When $D$ is understood we will omit it.

We now state a modified version of McDiarmid's theorem that holds for $\eta$-almost $\beta$- bounded difference functions.

**Lemma A.2.** *Let $X_1, \ldots, X_m$ be i.i.d. random variables. If $f$ satisfies $\eta$-almost $\beta$- bounded difference and takes values between $0$ and $M$, then,*

$$Pr\{f(X_1, \ldots, X_m) - E[f(X_1, \ldots, X_m)] \geq \epsilon\} \leq \exp\left[\frac{-2\epsilon^2}{m(\beta + M\eta)^2}\right] + \eta.$$

Lemma $A.2$ follows directly from a result shown in (Combes, 2015, Proposition 2). Since the proof is available in Combes (2015) we omit it here.

| Symbol | Explanation |
|--------|-------------|
| $L_S$ | Worst Case Training Gradient. |
| $\sigma_S$ | Second Moment of Step Training Gradient. |
| $L_g$ | Test Gradient. |
| $K_S$ | Training Smoothness Constant. |
| $\mathcal{X}, \mathcal{Y}$ | Input and output Space. |
| $D$ | Distribution over $\mathcal{Z} \subset \mathcal{X}\mathcal{Y}$. |
| $z = (x_z, y_z)$ | Input point and label picked from distribution $D$ defined over $\mathcal{Z}$. |
| $r \in \mathcal{R}$ | random string from a random set to show randomness in an algorithm. |
| $S$ | Training set of size $m$. |
| $S^i$ | Training set $S$ with $i^{th}$ point replaced by another point picked i.i.d from $D$. |
| $A_{S,r}$ | Training Algorithm. |
| $\ell(\cdot, \cdot)$ | Bounded value Loss function with domain $\mathcal{Y} \times \mathcal{Y} \to [0, M]$. |
| $R(A_{S,r})$ | Risk (Population error). |
| $R_e(A_{S,r})$ | Empirical Risk (Training error). |
| $\boldsymbol{w}_t$ | Weight of the parameterized function trained by SGD at $t^{th}$ step. |
| $\alpha_t$ | Learning rate at $t$-th step of SGD. |
| $\|.\|_\sigma$ | Spectral norm of matrix. |

| Symbol | Explanation |
|--------|-------------|
| $\tau$ | Total number of epochs, each epoch is of $m$ step, $\boldsymbol{w}_0$ is the initial weight. |
| $\pi \in \Pi$ | $\pi$ is some permutation of $m$ points picked from set of all possible permutation $\Pi$. |
| $r_{init}$ | Random initialization i.e. $\boldsymbol{w}_0$. |
| $r_p$ | A random permutation for $m$ points. |
| $r = (r_{init}, r_p)$ | Random string $r$ having $\boldsymbol{w}_0$ and $\{\pi_i\}_{i=0}^{\tau-1}$ i.e., for all epochs. |
| $L_l$ | Local parameter Lipschitz constant. |
| $K_l$ | Local parameter Smoothness constant. |
| $W^l$ | Weight matrix of $l$-th layer on NN. |
| $W_{j,:}^l$ | The $j$-th row of $l$-th layer weight of NN. |
| $w_{i,j}^{(l)}$ | Weight value of $l$-th layer from $i^{th}$ neuron of $l$-th layer to $j$-th neuron of $l+1$-th layer. |
| $\mathcal{T}$ | Size of test set. |
| $f(\boldsymbol{w}_t, z)$ | Loss at $t$-th step of SGD computed on point $z$. |
| $f(r, S, z)$ | Loss of NN trained on set $S$ and evaluated on point $z$. |

Table 3: Notation used in the body of the paper.

# B  Almost Sure (a.s.) Support Stability of SGD Proved

*Proof of Lemma 4.3.* Let $f$ be the partial function of $\boldsymbol{w}$ (i.e., assuming $z$ is already given) is locally Lipschitz at $\boldsymbol{w} \in A$, there is an $\varepsilon_{\boldsymbol{w}} > 0$ and an $L_{\boldsymbol{w}} > 0$ such that for all $\boldsymbol{w}' \in \mathbb{R}^n$ with $\|\boldsymbol{w} - \boldsymbol{w}'\| \leq \varepsilon_{\boldsymbol{w}}$, $|f(\boldsymbol{w}) - f(\boldsymbol{w}')| \leq L_{\boldsymbol{w}}\|\boldsymbol{w} - \boldsymbol{w}'\|$. So, let us turn our attention to those $\boldsymbol{w}' \in A$ that lie outside the ball of radius $\varepsilon_{\boldsymbol{w}}$ around $\boldsymbol{w}$. Note that for such a $\boldsymbol{w}'$, if $B > 0$ is the bound on $f$, we have that

$$\frac{|f(\boldsymbol{w}) - f(\boldsymbol{w}')|}{\|\boldsymbol{w} - \boldsymbol{w}'\|} \leq \frac{2B}{\varepsilon_{\boldsymbol{w}}}.$$

Therefore the "global" Lipschitz constant for $f$ within $A$ is $\max\{L_{\boldsymbol{w}}, 2B/\varepsilon_{\boldsymbol{w}} : \boldsymbol{w} \in A\}$ which is bounded since $A$ is finite. This is valid for all the partial functions (i.e., for all $z \in \Omega$) and hence proves the theorem. $\qquad\square$

*Proof of Theorem 4.5.* For some $i \in [m]$ we couple the trajectory of SGD on $S$ and $S^i$ where $z_i \in S$ has been replaced with $z_i'$. Our random decision string $r$, in this case, is a random choice of an initial parameter vector, $\boldsymbol{w}_0$, and a random set of $\tau$ i.i.d permutations $\pi_0, \ldots, \pi_{\tau-1}$ of $[m]$ chosen uniformly at random. We use these random choices for training both the algorithms with $S$ and $S^i$. For $0 \leq j \leq \tau - 1$, we denote $\pi_j^{-1}(i)$ by $I_j$, i.e., $I_j$ is the (random) position where the $i$th training point is encountered in the $j$th training epoch. The key quantity we will track through the coupled training process will be

$$\delta_t = \|\boldsymbol{w}_t - \boldsymbol{w}_t'\|,$$

for $1 \leq t \leq \tau m$. If we can show that $\mathrm{E}_r[L_g \delta_{\tau m}]$ is bounded by some quantity $B$ almost surely, we can invoke the fact that $f$ is a.s. $L_l$-LPL to say that $\|\mathrm{E}_r[f(\boldsymbol{w}_t, z) - f(\boldsymbol{w}_t', z)]\| \leq \mathrm{E}_r[L_g \delta_{\tau m}] \leq B$ for all $z \in \mathrm{supp}\,(D)$, where $L_g$ is the Test Gradient.

We argue differently for the first epoch and differently for later epochs. For the first epoch, we note that for $t \leq I_0$, $\delta_t = 0$ since SGD performs identical moves in both cases. At $t = I_0 + 1$

$$\delta_{I_0+1} = \|\boldsymbol{w}_{I_0} - \alpha_{I_0}\nabla f(\boldsymbol{w}_{I_0}, z_i) - (\boldsymbol{w}'_{I_0} - \alpha_{I_0}\nabla f(\boldsymbol{w}'_{I_0}, z'_i))\| = \alpha_{I_0}\|\nabla f(\boldsymbol{w}_{I_0}, z_i) - \nabla f(\boldsymbol{w}'_{I_0}, z'_i)\|, \qquad (10)$$

where the second equality follows from the fact that $\boldsymbol{w}_{I_0} = \boldsymbol{w}'_{I_0}$ by the definition of $I_0$. Using Lemma 4.3 we can say that $\delta_{I_0+1} \leq 2\alpha_{I_0}L_S$ almost surely. Notice here we use data-dependent Worst Case Training Gradient $L_S$ which is only defined for points in set $S$, unlike Test Gradient. Now,

$$\delta_{I_0+2} \leq \|\boldsymbol{w}_{I_0+1} - \boldsymbol{w}'_{I_0+1}\| + \alpha_{I_0+1}\|\nabla f(\boldsymbol{w}_{I_0+1}, z_i) - \nabla f(\boldsymbol{w}'_{I_0+1}, z'_i))\|.$$

Here although the parameter vectors $\boldsymbol{w}_{I_0+1}$ and $\boldsymbol{w}'_{I_0+1}$ are not the same, $z_{\pi_0(I_0+1)}$ and $z'_{\pi_0(I_0+1)}$ are the same by the definition of $I_0$ (assuming that $I_0 \neq m$). Therefore we get that

$$\delta_{I_0+2} \leq \delta_{I_0+1} + \alpha_{I_0+1}K_S\delta_{I_0+1}$$

with probability 1 since from Lemma 4.3 we have that $f$ has a "global" smoothness property for the entire set of at most $2\tau m$ parameter vectors that will be encountered during the coupled training of $S$ and $S^i$. So we used Training Smoothness Constant ($K_S$). Noting that a similar recursion can be applied all the way to the end of the first epoch, i.e. till $t = m$ we get

$$\delta_m \leq 2\alpha_{I_0}L_S \prod_{t=I_0+1}^{m}(1 + \alpha_t K_S) \leq 2\alpha_{I_0}L_S \exp\left\{\sum_{t=I_0+1}^{m}\alpha_t K_S\right\}, \qquad (11)$$

with probability 1. Moving on to the next epoch, we note that we can make the argument above till the next point where the two training sequences differ, i.e., till the $m + I_1 + 1$st step. At this point we have,

$$\delta_{m+I_1+1} \leq \delta_{m+I_1} + \alpha_{m+I_1}\|\nabla f(\boldsymbol{w}_{m+I_1}, z_i) - \nabla f(\boldsymbol{w}'_{m+I_1}, z'_i))\|.$$

Since neither the parameter vector nor the training points are the same in the second term, we have no option but to use the data-dependent Worst Case Training Gradient to say that,

$$\delta_{m+I_1+1} \leq \delta_{m+I_1} + \alpha_{m+I_1}2L_S.$$

Since $\alpha_{m+I_1} < \alpha_{I_0}$, observing that our current bound for $\delta_{m+I_1}$ is larger than $\alpha_{m+I_1}2L_S$. Therefore

$$\delta_{m+I_1+1} \leq 2\delta_{m+I_1}.$$

So, we see that in the second and subsequent epochs, for time step $jm + I_j + 1$, $1 \leq j < \tau$ we have the bound

$$\delta_{jm+I_j+1} \leq 2\delta_{jm+I_j},$$

and for all $t > m + I_1, t \neq I_1, \ldots, I_{\tau-1}$ we have, as before, by the smoothness property that

$$\delta_{t+1} \leq \delta_t(1 + \alpha_{t+1}K_S).$$

Therefore, we have that

$$\delta_{\tau m} \leq 2\alpha_{I_0}L_S(2)^{\tau-1}\exp\left\{\sum_{t=I_0+1}^{\tau m}\alpha_t K_S\right\} \leq \alpha_0 L_S 2^\tau \frac{1}{I_0^{1-\rho(\tau,m)}}\exp\left\{\frac{\alpha_0 K_S\left((\tau m)^{\rho(\tau,m)} - I_0^{\rho(\tau,m)}\right)}{\rho(\tau,m)}\right\}. \qquad (12)$$

where, in the first inequality for ease of calculation we have retained the terms of the form $(1 + \alpha_{I_j}K_S)$, $2 \leq j < \tau$ in the product on the right although we can ignore them. In the second inequality, we have substituted $\alpha_t = \alpha_0/t^{(1-\rho(\tau,m))}$ and bound the summation using integration.

Finally, in order to compute $\mathrm{E}_r[L_g\delta_T]$ remember there were two source of randomness first is random initialization $\boldsymbol{w}_0$ or lets call it $r_{init}$ and random permutation $\pi$ lets call it $r_p$. Now because $r_{init}$ and $r_p$ are

independent we can write $\mathrm{E}_r \left[ L_g \delta_{\tau m} \right] = \mathrm{E}_{r_{init}} \left[ \mathrm{E}_{r_p} \left[ L_g \delta_{\tau m} | r_{init} \right] \right]$. Now in order to compute $\mathrm{E}_{r_p} \left[ L_g \delta_{\tau m} | r_{init} \right]$ note that $L_g, L_S$ and $K_S$ are constant.

Note that, since $\pi_0$ is uniformly drawn from the set of permutations of $[m]$, $I_0$ is uniformly distributed on $[m]$. Summing up the last term of (12) over $I_0 \in [m]$ and dividing further by $m$ we get

$$\mathrm{E}_{r_p} \left[ L_g \delta_{\tau m} | r_{init} \right] \le 2^\tau \alpha_0 L_g L_S \times \frac{1}{m} \sum_{I_0=1}^m \frac{1}{I_0^{1-\rho(\tau,m)}} \exp \left\{ \frac{\alpha_0 K_S \left( (\tau m)^{\rho(\tau,m)} - I_0^{\rho(\tau,m)} \right)}{\rho(\tau,m)} \right\}$$

Using integration we bound the summation part and also using $\exp(-\alpha_0 K_S / \rho(\tau,m)) \le 1$ we get the upper bound for the summation part as

$$\le U \left( \alpha_0, K_S, \rho(\tau,m) \right) \cdot \exp \left( \frac{\alpha_0 K_S}{\rho(\tau,m)} (\tau m)^{\rho(\tau,m)} \right)$$

where

$$U \left( \alpha_0, K_S, \rho(\tau,m) \right) := 1 + \frac{1 - \exp(-\alpha_0 K_S m^{\rho(\tau,m)} / \rho(\tau,m))}{\alpha_0 K_S},$$

we get

$$\mathrm{E}_{r_p} \left[ L_g \delta_T | r_{init} \right] \le 2^\tau \alpha_0 L_g L_S \cdot U \left( \alpha_0, K_S, \rho(\tau,m) \right) \cdot \frac{\exp \left\{ \frac{\alpha_0 K_S}{\rho(\tau,m)} (\tau m)^{\rho(\tau,m)} \right\}}{m} \tag{13}$$

taking $\rho(\tau,m) = \frac{\log \log m}{\log \tau + \log m}$ and expectation over $r_{init}$ we get the desired result. $\qquad\square$

We also present the proof of Theorem 4.7, which is very similar to the proof of Theorem 4.5 with some changes.

*Proof of Theorem 4.7.* The proof of this is very similar to the proof of Theorem 4.5. The main point is that in the paragraph after Equation 10, we can actually use the exact gradient maxed over all permutation at $I_0$-th step (lets call it $\sigma_{S,I_0}$) instead of $L_S$ as it's the gradient at $I_0$-th step. The rest of the steps follow similarly till equation 11 so we get

$$\delta_m \le 2\alpha_{I_0} \sigma_{S,I_0} \exp \left\{ \sum_{t=I_0+1}^m \alpha_t K_S \right\}$$

Now calculating $\mathrm{E}_r \left[ L_g \delta_m \right]$, note that earlier (in line 622) $L_S$ was constant w.r.t the permutation $(r_p)$ but here $\sigma_{S,I_0}$ depends on the step, so taking expectation over permutation is equivalent to taking expectation over random variable $I_0$ which is picked uniformly from 1 to $m$.

$$\mathrm{E}_{r_p} \left[ L_g \delta_m | r_{init} \right] \le \alpha_0 L_g \cdot \mathrm{E}_{r_p} \left[ \frac{\sigma_{S,I_0}}{I_0^{1-\frac{\log \log m}{\log m}}} \exp \left\{ \sum_{t=I_0+1}^m \alpha_t K_S \right\} \right]$$

So using Cauchy Schwarz inequality to separate expectation over the random variable $\sigma_{S,I_0}$

$$\mathrm{E}_{r_p} \left[ L_g \delta_m | r_{init} \right] \le \alpha_0 L_g \cdot \sqrt{\mathrm{E}_{r_p} \left[ \sigma_{S,I_0}^2 \right]} \sqrt{\frac{1}{m} \sum_{I_0=1}^m \frac{1}{I_0^{2-\frac{2\log \log m}{\log m}}} \exp \left\{ \sum_{t=I_0+1}^m 2\alpha_t K_S \right\}}$$

Upper bounding summation inside exponent by integration exactly like we did in equation 12. we get the second square-root terms as

$$\le \frac{1}{m} \sum_{I_0=1}^m \frac{1}{I_0^{2-\frac{2\log \log m}{\log m}}} \exp \left\{ \frac{2\alpha_0 K_S \left( (m)^{\frac{\log \log m}{\log m}} - I_0^{\frac{\log \log m}{\log m}} \right) \log m}{\log \log m} \right\}$$

Assuming $p = \frac{\log\log m}{\log m}$ to simplify the equation,

$$\leq \frac{1}{m}\exp\left\{\frac{-2\alpha_0 K_S}{p}\right\}\left(\exp\frac{-2\alpha_0 K_S}{p} + \sum_{I_0=2}^{m}\frac{1}{I_0^{(2-2p)}}\exp\left\{\frac{-2\alpha_0 K_S I_0^p}{p}\right\}\right)$$

Now, we use integration to bound the summation term,

$$\sum_{I_0=2}^{m}\frac{1}{I_0^{(2-2p)}}\exp\left\{\frac{-2\alpha_0 K_S I_0^p}{p}\right\} = \int_{x=1}^{m} x^{-2(1-p)}\exp\left\{\frac{-2\alpha_0 K_S}{p}x^p\right\}$$

Substituting $u = \frac{x^p}{p}$, and in next step upper bounding $(u\cdot p)^{(1-1/p)}$ by 1 (as $p < 1$ and $u\cdot \geq 1$ throughout the limit) and then putting the limits.

$$\sum_{I_0=2}^{m}\frac{1}{I_0^{(2-2p)}}\exp\left\{\frac{-2\alpha_0 K_S I_0^p}{p}\right\} = \int_{u=\frac{1}{p}}^{m^p/p}(u\cdot p)^{(1-1/p)}\exp\left\{-2\alpha_0 K_S u\right\}$$

$$\leq p^{1-1/p}\int_{u=1/p}^{m^p/p}\exp\left\{-2\alpha_0 K_S u\right\}$$

$$\leq \frac{p^{1-1/p}}{2\alpha_0 K_S}\exp\left\{\frac{-2\alpha_0 K_S}{p}\right\}$$

Using this and putting the value of $p$ and taking $p^{1-1/p}$ we get,

$$\mathrm{E}_{r_p}\left[L_g\delta_m|r_{init}\right] \leq \alpha_0 L_g\cdot\sqrt{U'(\alpha_0, K_S, m)}\sqrt{\mathrm{E}_{r_p}\left[\sigma_{S,I_0}^2\right]}\cdot\frac{1}{m^{\frac{1}{2}-\frac{\alpha_0 K_S(\log m-1)}{\log\log m}}} \tag{14}$$

where

$$U'(\alpha_0, K_S, m) = 1 + \frac{1}{2\alpha_0 K_S}$$

and expectation over $r_p$ for $\sigma_{S,I_0}$ is just taking the average across SGD steps, so we get $\sigma_S$. $\qquad\square$

We also present the proof of Corollary!4.10.

*Proof of Corollary 4.10.* Writing assumption P1 averaged over $m$ steps and multiplying both side by $\alpha_0$ we get,

$$\frac{1}{m}\sum_{t=1}^{m}\alpha_0\left[\frac{1}{m}\sum_{z\in S}\|\nabla f(\boldsymbol{w}_t, z)\|^2\right] \leq \alpha_0 M + \frac{\alpha_0 M_G}{m}\sum_{t=1}^{m}\left\|\frac{1}{m}\sum_{z\in S}\nabla f(\boldsymbol{w}_t, z)\right\|^2$$

Now we take total expectation over the above equation and use Theorem 4.9 to bound the R.H.S of the equation. For L.H.S first, let the worst gradient (because of permutation) at $t$-th step computed on point $z_i$ is $L_{z_i,t} = \max_{\boldsymbol{w}\in A_t}\{\frac{\partial}{\partial\boldsymbol{w}}f(\boldsymbol{w}, z_i)\}$, where set $A_t = \cup_{\pi_0\in\Pi}\{\boldsymbol{w}_{t,\pi_0}\}$. Keeping this in mind, we can write L.H.S as,

$$\text{L.H.S} = \mathrm{E}\left[\frac{1}{m}\sum_{t=1}^{m}\left[\frac{1}{m}\sum_{i=1}^{m}L_{z_i,t}^2\right]\right] \tag{15}$$

$$= \mathrm{E}\left[\frac{1}{m}\sum_{t=1}^{m}\left[\frac{1}{m}\sum_{i=1}^{m}\mathrm{E}_{z_i\sim\mathcal{Z}}\left[L_{z_i,t}^2\right]\right]\right] \tag{16}$$

$$\tag{17}$$

Now note that because all $z_i$ are identical (picked from the same distribution) so expectation over them will be equal. So $\max_{i\in[m]}(\mathrm{E}_{z_i\sim\mathcal{Z}}\left[L_{z_i,t}^2\right]) = \mathrm{E}_{z_i\sim\mathcal{Z}}\left[L_{z_i,t}^2\right]$ and therefore we can replace the above with SMSTrG (i.e. $\sigma_S$). Using this, we get the statement of Corollary 4.10. $\qquad\square$

# C   Neural Networks: Characterization and Proofs

| Symbol | Explanation |
|---|---|
| $k_l$ | Number of neurons in $l$-th layer. |
| $H$ | Depth of neural network. |
| $\mathsf{in}_{i,j}(\boldsymbol{w}, x)$ | Input to the $i$-th neuron of $j$-th layer. |
| $\mathsf{out}_{i,j}(\boldsymbol{w}, x)$ | Output of the $i$-th neuron of $j$-th layer. |
| $\mathsf{out}_{H,1} = out(\boldsymbol{w}, x)$ | Is the label returned by the neural network. |
| $\phi(\boldsymbol{w}, x)$ | Polynomial associated with fully connected NN. |
| $\phi_{i,j}(\boldsymbol{w}, x)$ | Base polynomial associated with $j$-th neuron of $i$-th layer. |
| $G_{\boldsymbol{w}, x}$ | The set of weights need to set to zero, to apply all closed ReLU gate in NN (with ReLU). |
| $\phi(\boldsymbol{w}, x)\{G_{\boldsymbol{w}, x}\}$ | A neural network with ReLU activation with closed ReLU gates set to 0. |
| $\mathsf{lab}(x)$ | Label of point $x$. |
| $I_j$ | Position in permutation in $j$-th epoch when $i$-th training points (i.e. the replaced point) is encountered based on $\pi_j$. |
| $\delta_t$ | Norm of difference between weights for $S$ and $S^i$ at $t$-th step of SGD (i.e., $\|\boldsymbol{w}_t - \boldsymbol{w}'_t\|$). |

Table 4: Notation used in section C

In order to prove Theorem 5.2 we first need to describe a characterization of neural networks that allows us to get a better insight into their smoothness properties. We present the characterization in Section C.1 and the proof in Section C.2.

After that, we continue presenting the proof of Section 5. In Sections C.2 we also present the proof of Theorem 5.1, Proposition 5.3 and Theorem 5.4. Then finally in Section C.3 we present the proofs of Section 5.2

## C.1   A Polynomial-based Characterization Neural Networks

### C.1.1   Neural Network Terminology

Neural networks provide a family of parameterized functions of the form we have discussed in Section 4. The parameter vector $\boldsymbol{w} \in \mathbb{R}^n$ is applied over a network structure with layers. In this case, we specify $\mathcal{Z}$ to be $\mathbb{R}^d \times \mathbb{R}$, i.e., the data points are from $\mathbb{R}^d$ and the label is from $\mathbb{R}$, i.e., the NN has a single output. We will denote the depth of the network by $H$. The layers will be numbered 0 to $H$ with layer 0 being the *input layer*. The number of neurons in layer $i$ will be $k_i$. For this discussion, we assume a fully connected network. We will denote by $w^i_{j,k}$ the weight of the edge from the $j$ neuron of the $i$th layer to the $k$th neuron of the $i + 1$st layer. For the NN with parameters $\boldsymbol{w}$ at a point $x \in \mathbb{R}^d$ we will denote the input into the $j$th neuron of the $i$th layer by $\mathsf{in}_{i,j}(\boldsymbol{w}, x)$ and its output by $\mathsf{out}_{i,j}(\boldsymbol{w}, x)$. Further, we will assume that all neurons in all layers of the network except the input layer and the output layer have ReLU activation applied to them. In case the output of a node is 0 due to ReLU activation we will say the ReLU gate is *closed* otherwise we will say it is *open*. The label output by the network will be $\mathsf{out}_{H,1} = \mathsf{out}(\boldsymbol{w}, x)$. For each exposition, we will assume that $\mathsf{out}(\boldsymbol{w}, x) = 1$ if $\mathsf{in}(\boldsymbol{w}, x) > 0$ and 0 otherwise, i.e., there are only two labels in $\mathcal{Y}$. For convenience we will denote this architecture as $\mathcal{N}$.

### C.1.2   Multivariate Polynomials Associated with a Neural Network

Given a set of indeterminates $x = x_1, \ldots, x_l$, let $\mathcal{P}(x)$ be the set of multivariate polynomials on $x_1, \ldots, x_l$ with real coefficients. For any polynomial $p(x)$, $i_1, \ldots, i_q \in [l]$ and any $\alpha_1, \ldots, \alpha_q \in \mathbb{R}$ for some $q \le l$, we will denote by $p(x)\left\{x_{i_j} = \alpha_j : j \in [q]\right\}$ the polynomial in $\mathcal{P}(x \setminus \{x_{i_1}, \ldots, x_{i_q}\})$ that is obtained by setting all occurrences of $x_{i_j}$ to $\alpha_j$ in $p(x)$. In particular, $p(x)\{x_i = 0\}$ is the polynomial $p(x)$ with all monomials containing $x_i$ removed, and $p(x)\{x_i = 1\}$ retains all the monomials of $p(x)$ but those monomials that contain $x_i$ appear without the term $x_i$.

Returning to NNs, let us consider two sets of indeterminates: $x = \{x_i : i \in [d]\}$ and $\boldsymbol{w} = \{w_{j,k}^{(i)} : 0 \leq i < H, 1 \leq j \leq k_i, 1 \leq k \leq k_{i+1}\}$ and $k_0 = d$. Let us consider $\mathcal{N}$ defined in Sec. C.1.1 and create a version of it that replaced the ReLU activation at each node with the identity activation function. We will call this the *identity version* of $\mathcal{N}$ and denote it $I(\mathcal{N})$. We will say that $I(\mathcal{N})$ has the following polynomial associated with it:

$$\phi(\boldsymbol{w}, x) = \sum_{j_0=1}^{k_0} \sum_{j_1=1}^{k_1} \cdots \sum_{j_{H-1}=1}^{k_{H-1}} x_{j_0} w_{j_0,j_1}^{(0)} w_{j_1,j_2}^{(1)} \cdots w_{j_{H-1},1}^{(H-1)}.$$

Note that the output layer has only one neuron. We will refer to this as the *base polynomial* of $\mathcal{N}$. The base polynomial associated with the $j$th neuron in layer $i$ can be derived from the base polynomial of the network, we express this in figure 4 and also write formally as follows

$$\phi_{i,j}(\boldsymbol{w}, x) = \frac{\phi(\boldsymbol{w}, x) \left\{w_{l_1,l_2}^{(i)}=0, w_{l_4,l_5}^{(l_3)}=1 : l_1 \in [k_i] \setminus \{j\}, l_2 \in [k_{i+1}], l_3 > i, l_4 \in [k_{l_3}], l_5 \in [k_{l_3+1}]\right\}}{\prod_{p=i+1}^{H} k_i}. \tag{18}$$

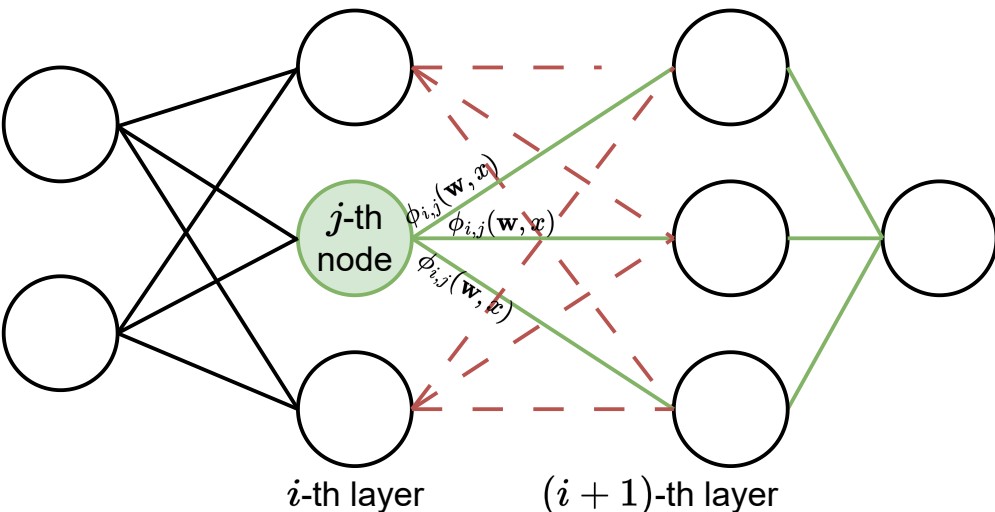

Figure 4: The output of the $j$-th neuron of the $i$-th layer represented represented by the base polynomial. Here the weights along the dotted red lines are set to zero and the weights along the green lines are set to one. The output of the neuron is represented by the values on the connection. Notice that the output is scaled by the product of the number of intermediate nodes because of which we divide it later in 18.

Also we could describe a Network whose say $i^{th}$ layer $j^{th}$ neuron's gate is closed by $\phi(\boldsymbol{w}, x)\{w_{l_1,j}^i = 0, \forall l_1 \in k_{i-1}\}$, This is represented by the figure 5. We will write $G_{\boldsymbol{w},x}$ as the set of weights needed to be equated to zero for all closed ReLU gates. It's clearly visible that due to ReLU activations varying at different points, there is no single polynomial that captures the output of the NN everywhere in $\mathbb{R}^n \times \mathbb{R}^d$. However, the following observation shows a way of defining polynomials that describe the output over certain subsets of space.

**Observation C.1.** *Given $\boldsymbol{w} \in \mathbb{R}^n$ and $x \neq (0, \ldots, 0) \in \mathbb{R}^d, i \in [H], j \in [k_i]$ and $\phi_{i,j}(\boldsymbol{w}, x)\{G_{\boldsymbol{w},x} = 0\}$ be the polynomial representing output and $G_{\boldsymbol{w},x}$ be the set of weights for closed ReLU gates as discussed above. For the case where $\mathsf{in}_{l_1,l_2}(\boldsymbol{w}, x) \neq 0$ for all $1 \leq l_1 \leq i$ and all $1 \leq l_2 \leq k_{l_1}$, there is an $\epsilon > 0$ depending on $\boldsymbol{w}, x$ such that, for all $\boldsymbol{w}'$ with $\|\boldsymbol{w} - \boldsymbol{w}'\| < \epsilon$,*

$$\phi_{i,j}(\boldsymbol{w}', x)\{G_{\boldsymbol{w}',x} = 0\} = \phi_{i,j}(\boldsymbol{w}', x)\{G_{\boldsymbol{w},x} = 0\}$$

*i.e. the polynomial remains same for $\boldsymbol{w}'$ and $\boldsymbol{w}$.*

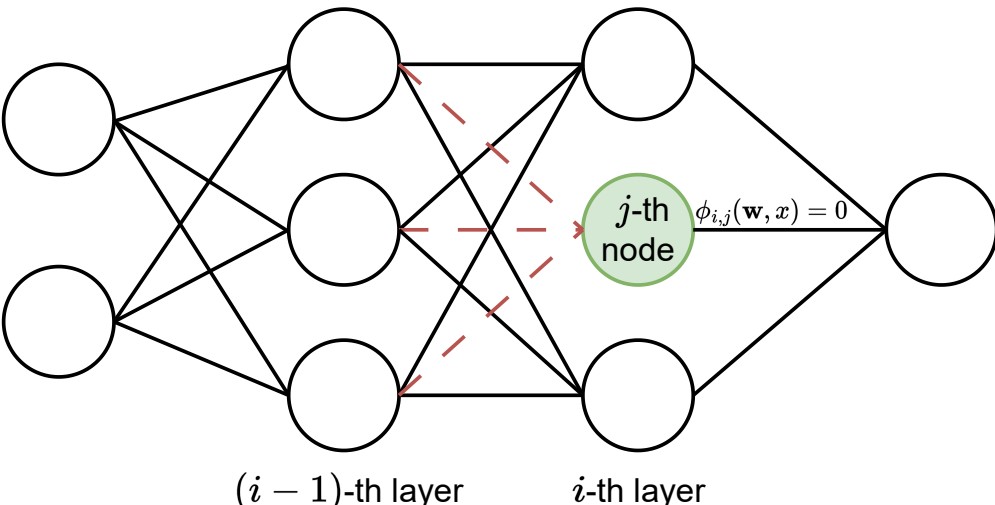

Figure 5: For a neural network with ReLU, if the $i$-th layers $j$-th neuron's ReLU gate is closed, this is represented by the base polynomial. Here the dotted red lines are set to zero.

*Proof.* Since $\mathsf{in}_{i,j}(\boldsymbol{w}, x)$ is strictly separated from 0 and there are only a finite number of neurons in the network there must be an $\epsilon$ small enough for which all open ReLU gates remain open and all closed gates remain closed. And because of this, we can use the same polynomial with new weights as no ReLU gate switches their state. $\qquad\square$

## C.2 Proofs for Section 5.1

We use the NN characterization for the proof of Theorem 5.2.

*Proof of Theorem 5.2.* The idea behind this proof is due to (Milne, 2019, Lemma 1) who used it for a different purpose. From Observation C.1 it follows that if we have $x \neq (0, \dots, 0) \in \mathbb{R}^d$ such that $\mathsf{in}_{i,j}(\boldsymbol{w}, x) \neq 0$ for all $1 \leq i \leq H$ and all $1 \leq j \leq k_i$, then $\mathsf{out}(\boldsymbol{w}, x)$ is, in fact, just the polynomial $\phi(\boldsymbol{w}', x)\{G_{\boldsymbol{w}, x} = 0\}$ within a small neighbourhood of $\boldsymbol{w}$. Therefore it is doubly differentiable. Since the loss function is also differentiable, we are done for all such values of $x$.

So now let us consider the set of points $x$ for which $i$ is the smallest layer index such that $\mathsf{in}_{i,j}(\boldsymbol{w}, x) = 0$. In case there are two such indices, we break ties using the neuron index $j$. By Observation C.1, in a neighbourhood of $\boldsymbol{w}$, $\mathsf{in}_{i,j}(\boldsymbol{w}, x)$ is a polynomial in $\boldsymbol{w}$ and $x$ for each $x$.

Now, we consider two cases. In the first case, $\mathsf{out}_{i-1,j'}(\boldsymbol{w}, x) = 0$ for all $j' \in [k_{i-1}]$, i.e., all the ReLU gates from the previous layers are closed because $\mathsf{in}_{i-1,j'}(\boldsymbol{w}, x) < 0$ for all $j' \in [k_{i-1}]$. In this case $\mathsf{out}(\boldsymbol{w}', x) = 0$ everywhere in the neighbourhood guaranteed by Observation C.1 and therefore $\ell(\mathsf{out}(\boldsymbol{w}', x), \mathsf{lab}(x))$ is doubly differentiable in the parameter space at $\boldsymbol{w}$ for all such $x$, where we assume that each data point has a label $\mathsf{lab}(x) \in \{0, 1\}$ associated with it. We note that this argument is easily portable to the case of a more general label set $\mathcal{Y}$ with the property described in the statement of Theorem 5.1 since $\mathsf{in}_{H,1}$ will be 0 everywhere in a small neighbourhood.

In the second case we have some $j' \in [k_{i-1}]$ such that $\mathsf{out}_{i-1,j'}(\boldsymbol{w}, x) > 0$. Let $C_{i,j} \subseteq \mathbb{R}^d$ be those $x$ for which this case holds. $C_{i,j}$ contains the solutions to $\mathsf{in}_{i,j}(\boldsymbol{w}, x) = 0$. Since we are working with a specific value of $\boldsymbol{w}$, this simply becomes a polynomial in $x$. In fact, inspecting the definition of base polynomials we note that when $\boldsymbol{w}$ is fixed $\mathsf{in}_{i,j}(\boldsymbol{w}, x)$ is simply a linear combination of $x_1, \dots, x_{\mathbb{R}}^d$. This implies that $C_{i,j}$ is a hyperplane in $\mathbb{R}^d$. We note that this argument can also be made of the output node under the condition on the label set given in the statement of Theorem 5.1 because for $\mathsf{in}_{H,1}(\boldsymbol{w}, x)$ to give a value that lies on the boundary between two sets with different labels for a given $\boldsymbol{w}, x$ must be drawn from a set of Lebesgue measure 0.

Since the network size is finite the set of all possible values of $x$ for which case 2 occurs, i.e., $\bigcup_{i \in [H], j \in [k_i]} C_{i,j}$ is a finite union of hyperplanes in $\mathbb{R}^d$ and therefore a set of Lebesgue measure 0. $\qquad \square$

*Proof of Proposition 5.3.* Let us consider the partial derivative w.r.t $w_{i,j}^{(l)}$. For this let $I_{i,j}^{(l)}, A_j^{(l+1)}$ and $B_i^{(l-1)}$ be 3 matrices of size $W^{(l)}, W^{(l+1)}$ and $W^{(l-1)}$ respectively such that $I_{i,j}^{(l)}[i,i] = 1$ and reset all entries are 0, $A_j^{(l+1)}[k,j] = W^{(l+1)}[k,j], \forall k$ and rest all entries are 0 and $B_i^{(l-1)}[i,k] = W^{(l-1)}[i,k], \forall k$ and rest all entries are one. Using these 3 matrices and the weight matrices we can compute the gradient as

$$\frac{\partial \phi(\boldsymbol{w}, x)}{\partial w_{i,j}^{(l)}} = W^{(H)} \cdots W^{(l+2)} \cdot A_j^{(l+1)} \cdot I_{i,j}^{(l)} \cdot B_i^{(l-1)} \cdot W^{l-2} \cdots W^1 \cdot x \tag{19}$$

Let $M'(l, i, j)$ be a matrix such that

$$M'_{l,i,j} = A_j^{(l+1)} \cdot I_{i,j}^{(l)} \cdot B_i^{(l-1)}$$

Although we have scalar values taking spectral norm on both sides of eq 19 we get

$$\left| \frac{\partial \phi(\boldsymbol{w}, x)}{\partial w_{i,j}^{(l)}} \right| = \prod_{k=1}^{H} \|W^{(k)}\|_\sigma \frac{\|M'_{l,i,j}\|_\sigma}{\|W^{(l+1)}\|_\sigma \cdot \|W^{(l)}\|_\sigma \cdot \|W^{(l-1)}\|_\sigma} \|x\|$$

Now lets define another matrix $M_l$ such that $(p, q)^{th}$ element of matrix $M_l[p, q] = \|M'_{l,i,j}\|_\sigma$. Now, the expression for $2, 2$ norm (Frobenius norm) of the gradient vector directly gives us the required expression for bound on gradients.

We can give a similar argument for bounding $K_S$, for some $\boldsymbol{w}_{i_1,j_1}^{(l_1)}$ and $\boldsymbol{w}_{i_2,j_2}^{(l_2)}$ we have

$$\left| \frac{\partial^2 \phi(\boldsymbol{w}, x)}{\partial w_{i_2,j_2}^{(l_2)} \partial w_{i_1,j_1}^{(l_1)}} \right| \leq \prod_{k=1}^{H} \|W^{(k)}\|_\sigma \left( \frac{\|M'_{l_1,i_1,j_1}\|_\sigma}{\|W^{(l_1+1)}\|_\sigma \cdot \|W^{(l_1)}\|_\sigma \cdot \|W^{(l_1-1)}\|_\sigma} \right)$$
$$\cdot \left( \frac{\|M'_{l_2,i_2,j_2}\|_\sigma}{\|W^{(l_2+1)}\|_\sigma \cdot \|W^{(l_2)}\|_\sigma \cdot \|W^{(l_2-1)}\|_\sigma} \right) \|x\|$$

Note that the above equation is exactly if $l_1 + 2 < l_2$ or $l_1 - 2 > l_2$ and for the rest of the case we can use this as the upper bound this is because for a matrix $M$ spectral norm $\|M\|_\sigma$ is upper bound for when we set all except one row or column of matrix to zero and calculate the spectral norm. Now if we take the $2, 2$ norm (Frobenius norm) of the Hessian matrix we get the desired result. $\qquad \square$

We now show the proof of Theorem 5.4 present in the discussion of Section 5.1.2, where we bound the gradients encountered.

*Proof of Theorem 5.4.* We first write the $\ell_2$ norm of gradients of parameter vector $\boldsymbol{w}$ as,

$$\|\nabla_{\boldsymbol{w}} f(\boldsymbol{w}, x)\|_2^2 = \|\nabla_{W^1} f(\boldsymbol{w}, x)\|_2^2 + \|\nabla_{W^2} f(\boldsymbol{w}, x)\|_2^2$$

Calculating gradients norm for both layers, assuming for given $\boldsymbol{w}_0$ (i.e., weight at initialization, note that this is part of $r$ randomness), we have a value of weights bounded above by $B_1$

$$\left\| \frac{\partial f(\boldsymbol{w}, x)}{\partial w_j^{(2)}} \right\|_2^2 = |\langle W_{j,:}^1, x \rangle|^2, \text{ Where } W_{j,:}^1 \text{ are the } j^{th} \text{ row of } W^1$$
$$\leq B_1^2 \cdot D^2$$

The effective dimension of $x$ is $D$, so the above dot product dimension will also be bounded by $D$ as $x$ is $0$ in all other dimensions.

$$\left\| \frac{\partial f(\boldsymbol{w}, x)}{\partial w_{i,j}^{(1)}} \right\|_2^2 = |w_i^{(2)} x_j|^2$$
$$\leq B_1^2$$

So, summing up the squared partial derivatives across all parameters we get,

$$\|\nabla_{\boldsymbol{w}} f(\boldsymbol{w}, x)\|_2^2 \leq d_1 \cdot D \cdot B_1^2 + d_1 \cdot D^2 \cdot B_1^2$$
$$= B_1^2 \cdot d_1 \cdot D(1 + D)$$

$\square$

## C.3  Proofs for Section 5.2

*Proof of Theorem 5.6.* First lets assume that assumption P1 holds so we need to show $\mathrm{E}_{\boldsymbol{w}_0, z \in S} [f(\boldsymbol{w}_0, z)]$ is bounded, so calculating the value for this,

$$f(\boldsymbol{w}_0, z) = |y - O(\boldsymbol{w}, x)|$$
$$= \left| y - \sum_{i=1}^k w_i^{(2)} w_{i,1}^{(1)} x_1 - \sum_{i=1}^k w_i^{(2)} w_{i,2}^{(1)} x_2 \right|$$
$$= |y| + \left| \sum_{i=1}^k w_i^{(2)} w_{i,1}^{(1)} x_1 \right| + \left| \sum_{i=1}^k w_i^{(2)} w_{i,2}^{(1)} x_2 \right|$$

Let $o_1(\boldsymbol{w}_0) = \sum_{i=1}^k w_i^{(2)} w_{i,1}^{(1)}$ and $o_2(\boldsymbol{w}_0) = \sum_{i=1}^k w_i^{(2)} w_{i,2}^{(1)}$ for ease of writing, then take expectation over $\boldsymbol{w}_0$, we directly place $\mathrm{E}_{\boldsymbol{w}_0} [|o_1(\boldsymbol{w}_0)|] = \mathrm{E}_{\boldsymbol{w}_0} [|o_2(\boldsymbol{w}_0)|] \leq \frac{2k\sigma^2}{\pi}$ because of half normal distribution and i.i.d assumption ,i.e.,

$$\mathrm{E}_{\boldsymbol{w}_0} [f(\boldsymbol{w}_0, z)] \leq |y| + \frac{2k\sigma^2}{\pi} (|x_1| + |x_2|)$$

Taking expectation over $z$, we have

$$\mathrm{E}_{\boldsymbol{w}_0, z \in S} [f(\boldsymbol{w}_0, z)] = 1 + \frac{4k\sigma^2 \mu_p}{\pi} \tag{20}$$

Now, we show the assumption P1 holds. Note that we ignore the case when weights are exactly zero or weights become exactly equal to other weights to avoid zero in the denominator. We take $M = 0$ from assumption P1. Now we take the upper bound of L.H.S. of assumption P1 (without expectation),

$$\|\nabla f(\boldsymbol{w}, z)\|^2 \leq n \cdot \max_j (\nabla f(\boldsymbol{w}, z)_j^2)$$

and we take the lower bound of R.H.S. using

$$n \cdot \min_i (\mathrm{E}_{z \in S} [\nabla f(\boldsymbol{w}, z)_i])^2 \leq \|\mathrm{E}_{z \in S} [\nabla f(\boldsymbol{w}, z)]\|^2$$

Using the above two inequalities and taking $M = 0$ in assumption P1, we get,

$$\left( \frac{\mathrm{E}_{z \in S} \left[ \max_{j \in [n]} \{ \nabla f(\boldsymbol{w}, z)_j^2 \} \right]}{\min_{i \in n} \{ \mathrm{E}_{z \in S} [\nabla f(\boldsymbol{w}, z)_i]^2 \}} \right) \leq M_G$$

Now, calculating for numerator, we first write the max over the square of gradients,

$$\max_j \{(\nabla f(\boldsymbol{w}, z)_j)^2\} = \max_j \left\{ \left( \frac{\partial O(\boldsymbol{w}, x)}{\partial w_{j_2, j_3}^{(j_1)}} \right)^2 \right\}$$

$$= \max\{(w_{j_2}^{(2)})x_{j_3}^2\}, \qquad \text{if } j_1 = 1$$

$$= \max\{(w_{j_2,1}^{(1)}x_1 + w_{j_2,2}^{(1)}x_2)^2\}, \quad \text{if } j_1 = 2$$

Let $w_h$ be the highest absolute value of weight(s) and $w_l$ be the lowest absolute value of weight(s). To easily calculate expectation, we take out the max weights across all, we get an upper bound for the numerator,

$$\mathrm{E}_{z \in S} \left[ \max_j \left\{ (\nabla f(\boldsymbol{w}, z)_j)^2 \right\} \right] \leq 2w_h^2 \sigma^2 \tag{21}$$

Now lower bounding denominator, so square of expectation of partial derivative, i.e.,

$$\mathrm{E}_{z \in S} \left[ \nabla f(\boldsymbol{w}, z)_i \right]^2 = \left( \mathrm{E}_{z \in S} \left[ \frac{\partial O(\boldsymbol{w}, x)}{\partial w_{i_2, i_3}^{(i_1)}} \right] \right)^2$$

$$= (w_{i_2}^{(2)} \mathrm{E}\,[x_{i_3}])^2, \qquad \text{if } i_1 = 1$$

$$= (w_{i_2,1}^{(1)} \mathrm{E}\,[x_1] + w_{i_2,2}^{(1)} \mathrm{E}\,[x_2])^2, \quad \text{if } i_1 = 2$$

For $i_1 = 2$ term, after taking expectation, we could write it as,

$$(w_{i_2,1}^{(1)} \mathrm{E}\,[x_1] + w_{i_2,2}^{(1)} \mathrm{E}\,[x_2])^2 = \frac{1}{4} \left( w_{i_2,1}^{(1)} + w_{i_2,1}^{(1)} \right)^2$$

Since we have $B$ for all ratios of weights we could use this to bound below the absolute difference between any pair of weights (i.e., $|w_l' - w_l| \geq |w_l/B - w_l|$), and we get

$$(w_{i_2,1}^{(1)} \mathrm{E}\,[x_1] + w_{i_2,2}^{(1)} \mathrm{E}\,[x_2])^2 \geq w_l^2 (B-1)^2 / B^2$$

So we can bound the whole denominator by,

$$\min_i \left\{ (\mathrm{E}_{z \in S}\,[\nabla f(\boldsymbol{w}, z)_i])^2 \right\} \geq \frac{1}{4} \min \left\{ w_l^2, \frac{w_l^2 (B-1)^2}{B^2} \right\} \tag{22}$$

$$\geq \frac{w_l^2 (B-1)^2}{4B^2} \tag{23}$$

Using 21 and 22 we get,

$$M_G = \frac{8B^4 \sigma^2}{(B-1)^2} \tag{24}$$

And from 20 and 24 we get the theorem statement. □

