# OpenReview forum: "Data-Dependent Generalization Bounds for Neural Networks with ReLU"
_TMLR — Accepted by TMLR_

### Review · Reviewer_FJNX · 2023-09-17

**Summary Of Contributions:**

As indicated in the abstract, the authors "try to establish that one of the correct data dependent quantities to look at while trying to prove generalization bounds, even for overparameterized neural networks, are the gradients encountered by stochastic gradient descent while training the model. If these are small, then the model generalizes."

**Audience:**

No

**Claims And Evidence:**

No

**Requested Changes:**

- Define formally the meaning of "generalization" such that readers may evaluate the claims regarding it.

- Clarify the setting of this work by formulating explicitly the target problem, which I think it is neural network classification.

- Accordingly, clearly restrict the loss function (s) under consideration. Is it not the cross-entropy as surrogate loss for training?

- The role of the set $\mathcal{R}$ and its elements (decision strings) is unclear. Needs discussion/illustration for adding clarity.

- The meaning of "loss function" needs to be specified before defining risks and other things. At least some discussion of the typical loss functions that are considered, and that one of those is referred to when we read "loss function" in the sequel.

- The notations $x_z$ and $y_z$ are unnecessary. Simply replace $z \sim D$ with $(x,y) \sim D$ and then you can write $\ell(A_{S.r}(x),y)$ in the definition of risk. Similarly, replace $z \in S$ with $(x,y) \in S$ in the sums for the empirical risk.

- Reading sometimes $A_{S,r}$ and sometimes $A_S$ or $A$ caused confusion. Can the relation between these things be declared? Perhaps a formal definition of each of these things would help to clear the confusion.

- When declaring $S^{i}$, after the definition perhaps add comments to the effect that $S^{i}$ is formed by replacing the $i$th entry of $S$ with an independent copy (which is taken as the $i$th entry from the "second set of size $m$"). By the way, this reminds of the double sample argument, which goes back to classical literature on statistical learning, which might be good to cite.

- Definition 3.1: Poor choices of terminology ($\eta$-almost support stability, a.s. support stability) which don't add much clarity. I suggest reformulating these things. The way I see it, the main idea being defined is that of "support stability" corresponding to the displayed condition, which may hold with some probability $1-\eta$ or with probability 1 (almost surely). Then you could reformulate this definition writing "support stability $\beta$ with probability $1-\eta$" and "support stability $\beta$ almost surely" -- the latter formulations are way more clear in conveying the meaning of what's being defined.

- I don't think the notation $[m]$ was declared. Just declare it somewhere before this definition.

- If we choose a constant loss function $\ell \equiv 1$, then this satisfies support stability $\beta$ for any $\beta>0$? This perhaps shows one of the problems in not restricting the meaning of "loss function" from the start.

- Also I'd like to flag the restriction "$\forall z \in \mathrm{supp}(D)$" in this definition. It appears to contradict the statements "with probability at least $1-\eta$" and "almost surely" so this needs clarification.

- Actually, a few lines below it is stated that "This probability is defined over the random choices of $Z_1, . . . , Z_{2m}$" which then suggests that Definition 3.1 needs to be rewritten making this explicit. Since the distribution of the random sample (i.i.d. points) of size $2m$ is $D^{\otimes 2m}$, this would be written saying "$D^{\otimes 2m}$-probability $1-\eta$" and "$D^{\otimes 2m}$-almost surely"

- Still, it is a very strong requirement that the inequality holds $\forall z \in \mathrm{supp}(D)$ (and $\forall i \in [m]$). Could the authors give illustrative examples of cases for which it is possible to calculate or estimate this condition.

- Theorem 3.2: Could the authors comment on the constant $c>0$ please. The exact value of this constant can make all the difference between the bound being useful or it being useless.

- The meaning of "symmetric in distribution" needs to be specified (before Theorem 3.2).

- Definitions 4.1 and 4.2: I don't know what meaning to map to "Given a set $\Omega$ defined over $\mathcal{Z}$."

- Theorem 4.4: Reading "We are given a labelled data set $\mathcal{Z}$" is surprising. I thought $\mathcal{Z}$ was reserved for the space of all possible instances and labels, i.e. $\mathcal{Z} = \mathcal{X}\times\mathcal{Y}$; while a "labelled data set" should be a finite sequence in this space.

- The rest of the theorem statement is really (!) hard to parse. Could this be improved?

- Similar comment for Theorem 5.1.

- Note that Theorem 5.1 neglects many details of the neural network architecture. Does this mean that the theorem holds for any choice of architecture (e.g. any depth and and widths in the hidden layers) as long as the output layer is 1-dimensional?

- Figure 2: The obvious question coming to mind is what is being plotted for "generalizaion error" here. If we take the definition stated earlier on in the paper that "generalization error" stands for the gap between risk and empirical risk, then I would ask how the authors obtained the values plotted here. Perhaps the plotted quantity is a proxy for the generalization error. I any case, this needs explanation.

- Another observation regarding Figure 2: The bound values appear to be loose. Definitely they are not nearly the best bound values for neural network classifiers reported in the literature. This raises the question about what is the take-home message that readers could get from reading this paper. If not tightness of the bound values, then it must be something else, but currently unclear.

- Bottom of page 11, bullet about NTK: I could not make sense of what's written here. Please elaborate and clarify.

- I think "data-dependent" needs hyphenation (throughout the paper).

**Strengths And Weaknesses:**

The most outstanding weakness of this work is its writing style, unfortunately. Reading it has been so unpleasantly hard that I am motivated to focus on criticizing the submission, regardless of the positive aspects (strengths) it might have.

The write up is missing a clearly formulated target problem (neural network classification?) and explicitly formulated target question(s) corresponding to the problem (e.g. generalization). Note that targeting generalization requires explicit definitions such that it might be possible to judge the claims. For instance, I didn't see explicit description of the meaning of "generalization error" in the introduction. A short sentence saying that this is the gap between the expected (population) error and the empirical error rate would be sufficient to clarify what is meant, and better if a cross-reference to the formal definition in Section 3.1 is given, to avoid confusions.

I flag for criticism the opening sentence "The low generalization error of Deep neural networks is now a well known empirical result" - if the meaning of generalization error is as I just described above, then this claim is unfounded. It might be that the authors (or the authors of the cited paper) are trying to refer in this claim to the gap between test error rate and training error rate. This needs clarification.

There are many other weaknesses that need attention, see my requested changes below.

---

> ### Author Response · Authors · 2023-11-10
> **Rebuttal by Authors**
>
> We would like to thank the reviewers for their close reading and their detailed comments. We have made an attempt to address these comments as fully as possible. However, in the process our rebuttal document has become very long and will exceed the 5000 word limit that Openreview places on posts. So, we have placed the entire rebuttal at the end of the updated pdf. We request the reviewers to find our responses there.
>
> In the main paper we have marked the edited text in 2 colors. Red indicates that the changes are important and add some extra information/value to the paper. Blue indicates the changes are mainly to increase readability, and they do not affect the paper's claims.
> We request the reviewers to go through them and respond and help us further improve this work.
>
> Thanks.

---

### Review · Reviewer_d5U1 · 2023-10-25

**Summary Of Contributions:**

The authors study the generalization of Rectified Linear Unit (ReLU) neural networks (NN) trained via stochastic gradient descent, through the lens of stability theory. First, they introduce a weakened notion of stability that holds with a certain probability and use its variation holding with probability one to derive generalization bounds for any algorithm satisfying their stability notion. Then they define a notion of data-dependent Lipschitzness and smoothness properties, which are later shown to hold for fully connected networks with ReLU activation functions. Bounds are provided for the constants of Lipschitzeness and smoothness in the case of a ReLU NN. Finally, experiments are run to corroborate the theoretical findings on two datasets.

**Audience:**

Yes

**Broader Impact Concerns:**

This is a paper with solely theoretical claims. As such, its potential practical impacts largely depend on its applications. Thus, I do not see the need for any discussion of broader impacts.

**Claims And Evidence:**

Yes

**Requested Changes:**

I have tried to group the requested changes into different categories.

## To facilitate reading

- Unifying Lipschitzness and smoothness: Definitions 4.1 and 4.2 can be unified under the same definition of Lipschitzness of a vector-valued function. Then, local parameter-Lipschitzness and local parameter-smoothness can be stated as simple specializations of that definition to the function itself and to its gradient, respectively. The same holds for Lemma 4.3.

- Both adding some informal description of the mathematical quantities presented and adding a formal description for informal statements.

  - At the beginning of Section 3.2, the sentence "Given the set $S$, we construct $S_i$ via replacing the $i-$th element of $S$ by an independently generated element from $D$" would facilitate conveying the idea, along with the given formula.

  - In the discussion just after the statement of Lemma 4.3, the expressions "set of weights encountered during training over all possible permutations" and "set of final parameter vectors produced by SGD for each of the possible permutations" would be much clearer if a formal definition is provided for the set $A$ in each case. This definition can for example be only in the appendix, but I think it clarifies these quantities.

- Base polynomials

  - Adding interpretations:

    - Full NN: for example, "output of a network with the identity activation function, i.e. fully linear".

    - Specific neuron: obtained by setting... In this case, a figure illustrating the concerned neuron and the operation applied to obtain the polynomial can significantly quicken understanding.

  - provide references for the base polynomial (if any)

- Adding a notation table in the supplementary material

- Using the same letter to index layers, and neurons. I agree this is nitpicky, but I think it would facilitate reading more. For example, the index $l$ for layers (including $l_i$ where $i$ is any index).

- Specifying the exact source of inspiration for the proofs (precisely which results in the mentioned reference.)

- In the second point of the list of contributions, a number of epochs of $c\log m$ is mentioned, but there is no explanation on the nature of constant $c$ (*e.g.* a universal constant, a constant depending on some parameters ...). Alternatively, if it is just the $\log m$ growth rate that is to be highlighted, then for instance, writing "for a number of epochs proportional to $\log m$" solves the issue.

- In Theorem 4.4

  - $F(\tau)$ 's expression can be directly incorporated in the bound, *i.e.* without introducing $F(\tau)$.
- Avoiding long sentences, or adding commas at least:

  - Corollary 4.5: The first sentence is very long, and needs a comma after:

    - "*Theorem 4.4*"

    - "*w.r.t.* $m$": also the word "then" should be removed, as the structure is "For an algorithm verifying conditions A, B, ..., there is a constant ...".

    - $c\log m$ *epochs*

- In proposition 5.3, it is written "Note that this equation holds for both Training Lipschitz ...". While It is understandable that the meant equation is the one providing a bound on $L_g$, attributing a number to it would be clearer.

- In the Definition $A.1$ in the appendix, the bounded differences property is called $\beta-$Lipschitzness. However, Lipschitzness is a well-known property that is totally different from the bounded differences. This can be confusing for readers and I recommend calling it the bounded differences property.


## Notation and terms issues

- No definition of the term "symmetric in distribution" was given before its first use in Theorem 3.2.

- It should be indicated whether constant $c$ in Theorem 3.2 is universal, and if not, what quantities it depends on.

- In Definitions 4.1 and 4.2, $z$ is used in the Lipschitzness condition for the first time, without being introduced. Only later it is specified that the constant $L_l$ or $K_l$ depends on $z$. This can be fixed by introducing $z$ at the moment when $\bm w$ is introduced.

- The logarithm is denoted "$\log$" in all of the paper except for the expression of $\rho$. Homogeneity of notation is better.


## Other suggestions

- Constructing a simple theoretical example in which data-dependent Lipschitz and smoothness constants are small or moderately large, but absolute ones are too large or even infinite.
- Adding a discussion on the interaction between the two terms in the bound in Corollary 4.5. Indeed, the first term divided by the second yields $\cfrac{(\log m)^{\cfrac{3}{2}}}{m^{\epsilon-\cfrac{1}{2}}}$, which means that one gets the "usual" rate of $\tilde{O}\left(\cfrac{1}{\sqrt{m}}\right)$ whenever $\epsilon\geq\cfrac{1}{2}$ (where the tilde hides the logarithmic factors), and will be slower if $\epsilon<\cfrac{1}{2}$. It would be interesting to analyze this behavior and to give an intuition of the slow rate, and when would it be beneficial to slow it for example. The overall rate can also be written as $\tilde{O}\left(m^{-\min\left(\epsilon,\cfrac{1}{2}\right)}\right)$.
- In the proof of Proposition 5.3, the product of norms is used to bound the norm of products. I think it would have been fine to let the norm of the product of matrices without further bounding it. Indeed, besides the fact that it already only incorporates quantities that appear in the bound stated by the proposition, it would result in a much tighter bound.
- The McDiarmid inequality for functions satisfying the bounded differences with high probability can be related to the result of [1].
- In the discussion just after Theorem 5.1, it is written "then these values will be high and reflect a bad generalization", where "these values" refers to $L_S$ and $K_S$. While the right-hand side of the generalization inequality of the theorem vanishes as $L_S$ approaches $0$ (since $F(\tau)$ is proportional to $L_S$), I do not think that seeing it holds for $K_S$ is straightforward. Indeed,on the one hand, when $K_S$ tends to 0, $m^{1-\frac{\alpha_0K_S}{\rho(\tau,m)}}$ tends to $m$. On the other one, we have $U(\alpha_0, K_S, \rho(\tau,m)) = 1 + \cfrac{1-\exp(-\alpha_0K_Sm^{\rho(\tau,m)}/\rho(\tau,m))}{\alpha_0 K_S}$ converges to its upper bound $1+\cfrac{m^{\rho(\tau,m)}}{\rho(\tau,m)} $ as $K_S$ vanishes. However, since $\rho(\tau,m) = \cfrac{\log\log m}{\log m + \log \tau}$, we have $m^{\rho(\tau,m)} = \exp(\cfrac{\log m\log\log m}{\log m + \log \tau})$ behaves as $\log m$ as $m$ grows, hence the bound on $U$ behaves as $\log m$. In the end, we would have a generalization bound that behaves as $O(\cfrac{(\log m)^3}{m} + \sqrt{\cfrac{\log m}{m}}) = \tilde O(\sqrt{\cfrac{1}{m}})$ . Hence, I think a more detailed explanation should be given.

## Formatting

- The title of Section 3 should not start with "a.s.". Rather, for example, "Almost Sure (a.s.) Support Stability ..."

- At the end of Corollary 4.5, one cannot write "and $\in$" as the "$\in$" operator needs a left-hand side. Instead, one can either write in full "and belongs to (0,1)" or, more shortly, $\epsilon \in (c,1)$.


## Grammar

- At the beginning of section 4, the first part of the sentence "The parameter vector at step t being ..." does not have a conjugated verb. It should rather be "The paraöeter vector at step t is denoted ..."

- Section 4.1: "required" --> "needed" (to avoid repetition with "requirement").


## Typos

- In the related work section "our work falls in this category" should be changed to, for instance, "in which our work falls" or "a category in which our work falls".
- In Equation (7) of the appendix, the norm of x is to remove.
- After Equation (7) in the appendix, "M'(l,i,j)" should be in latex math mode.
- In Theorem 5.1, the "$\in$" symbol in "$S \in D^m$" should be $\sim$, as $D^m$ is the distribution of $m$ samples drawn independently from $D$.

### References

[1] Combes, Richard. "An extension of McDiarmid's inequality." *arXiv preprint arXiv:1511.05240* (2015).

**Strengths And Weaknesses:**

The paper is overall well written and easy to follow in general, but some modifications can make reading easier, especially in the appendix. I have read both the main and the supplementary material.

## Strengths

- The area of generalization of neural networks is of capital interest to the machine learning community, and it is good to investigate it.

- Data-depending smoothness and Lipschitzness constants

- Proofs are provided for all of the theoretical claims. They are generally well written. A clear effort is made to facilitate reading and to convey the guiding idea for each proof.

- A good experimental evaluation.


## Weaknesses

- A notion of stability holding with any probability is introduced, only for its particular case of holding with probability one to be used later. In this case, introducing the almost sure version directly makes the paper easier to follow.

- Unclarity in some parts:

  - The terms $L_l$-Lipschitzness and $K_l$ smoothness allude to constants $L_l$ and $K_l$, yet these constants might depend on $z$. Isn't it more convenient to consider $L_l$ and $K_l$ as functions rather than constants in this case ?
- Several issues in Theorem 4.4:

  - Error in a claim: from the proof of the theorem, we have $U(\alpha_0, K_S, \rho(\tau,m)) = 1 + \cfrac{1-\exp(-\alpha_0K_Sm^{\rho(\tau,m)}/\rho(\tau,m))}{\alpha_0 K_S} $, in which the second term converges to $\cfrac{m^{\rho(\tau,m)}}{\rho(\tau,m)}$ rather than to 0. Also, due to the concavity of the numerator of the second term, one can bound U by $1+\cfrac{m^{\rho(\tau,m)}}{\rho(\tau,m)}$ which is more informative than the bound in the paper, as it corresponds also to the limit for very small $\alpha_0$.

  - Unsubstantiated claim: Just after the statement of Theorem 4.4, in the "*Data dependence with Training Lipschitz constant and Test Lipschitz constant*", there is the statement "we expect that if the unknown distribution D has a low variance then the Lipschitz constants will be small". Although this is a conjecture, some justification of motivation for it should be provided, experimentally at least (for example, by plotting the data-dependent Lipschitz constant bounds against the variance of a data set.

- Unsubstiantiated claim and unclarity on the behaviour of $\beta$ after Corollary 4.5: it is mentioned that $\beta = o(\cfrac{1}{m})$ with the conditions of Theorem 4.4 and Corollary 4.5. However, applying that to Theorem 3.2, one would expect a behaviour that is $\sqrt{\cfrac{\log m}{m}}$ as the second term in the generalization bounds dominates in this case. However, the bound given in the corollary still keeps a term growing in $\cfrac{1}{m^{\epsilon}}$. Also, I could not find the justification for this asymptotic behaviour of $\beta$ in the appendix.

- Many improvements can be made in order to facilitate reading, please refer to the "Requested changes" section for details.

---

> ### Author Response · Authors · 2023-11-10
> **Rebuttal by Authors**
>
> We would like to thank the reviewers for their close reading and their detailed comments. We have made an attempt to address these comments as fully as possible. However, in the process our rebuttal document has become very long and will exceed the 5000 word limit that Openreview places on posts. So, we have placed the entire rebuttal at the end of the updated pdf. We request the reviewers to find our responses there.
>
> In the main paper we have marked the edited text in 2 colors. Red indicates that the changes are important and add some extra information/value to the paper. Blue indicates the changes are mainly to increase readability, and they do not affect the paper's claims.
> We request the reviewers to go through them and respond and help us further improve this work.
>
> Thanks.

---

> ### Comment · Reviewer_d5U1 · 2023-11-14
> **Response to the authors' rebuttal.**
>
> Dear authors,
>
> Thank you for the time you took to incorporate the different points raised in the reviews in the new version. I have read the new modifications and I think they add a substantial improvement on the clarity of the paper. With this being said, I have the following remarks.
>
> ## Typos in the newly added text:
>
> - Line 55: "stabilityusing" --> "stability using".
>
> - Line 134: "algorith" --> "algorithm"
>
> - Line 181: "misleaing" --> "misleading".
>
> - Line 394: "squared $l_2$" --> "squared $l_2$ norms".
>
>
> ## Other remarks
>
> - Line 39: It is mentioned that the results hold "for both classification and regression cases." However, assumption N2 requires that the output space is countably infinite, which does not agree with the regression setting. Can it be relaxed to incorporate the uncountable setting ? Otherwise, can the scope be restricted to classification ?
>
> - Line 162: The expression "with a certain probability" is more convient to the previous version that considers an arbitrary probability $1-\eta$ in my opinion. I would replace "with a certain probability" to "almost everywhere" or more precisely $D^{2m}-$ or $D^{\otimes 2m}-$ almost everywhere.., as in the review of Reviewer FJNX.
>
> - Line 169: $z \in D$ does not hold as $D$ is not a set. Either $z \in \mathcal Z$ or $z \sim D$.
>
> - Line 326: The $\eta-$almost $\beta-$bounded difference is mentioned. However, it is only defined in the appendix, so a remark (even between parentheses) referring to the appendix is necessary.
>
> - Line 400: It is mentioned that "we need bound on just the square of the expectation of each term." By the Cauchy-Schwarz inequality, we have $\mathbb{E}[L_S.L_g] \leq \sqrt{\mathbb{E}[L_S^2]\mathbb{E}[L_g^2]}$. Hence, I think it is rather "the expectation of the square (or the second moment) of each term".
>
> - In line 710, I appreciate adding the "NN with identity function" precision to facilitate the interpreation of the base polynomial. However, as I read it "a fully connected NN defined in ..., i.e. NN with the identity function", this implies that the neural network defined in section D.1.1. has an identity activation function which is not the case. Also, the formulation makes it look as if the base polynomial is defined for an NN with an identity activation function, whereas it can be defined as the output for any fully connected NN **if we assume** it has the identity activation, while its weights are kept.
>
> - In my previous review, in the comment about the sources for inspiration in the proofs, I was meaning for example "[Reference R, Theorem T]" instead of just "[Refernce R]". It is not necessary but its role is to locate the result quickly in the mentioned references. I apologize for any lack of clarity.
>
>
> ## Questions:
>
> - In line 251, in assumption S1, it is written: "$L_l$-LPL w.r.t. $\operatorname{supp}(D)$ and "$K_l$-LPS w.r.t. $D$". Is there a difference between saying "w.r.t. $D$" or $\operatorname{supp}(D)$ ?
>
> - Is it possible to develop more on the advantages on the product of norms in Proposition 5.3, in terms of interpretability ?

---

> ### Author Response · Authors · 2023-11-21
> **Response to Post Rebuttal questions by Reviewer**
>
> Thank you for your response. We have updated the draft and marked recent changes with magenta. We are posting replies to your comment here and have also maintained it in the latest draft.
>
> > Typos in the newly added text
>
> We have fixed them now. Thank you.
>
> ---
> ## Other remarks
>
> > Line 39: It is mentioned that the results hold "for both classification and regression cases." However, assumption N2 requires that the output space is countably infinite, which does not agree with the regression setting. Can it be relaxed to incorporate the uncountable setting? Otherwise, can the scope be restricted to classification?
>
> The reviewer is right. We have now clarified on Line 39 that the scope is restricted to classification.
>
> ---
>
> > Line 162: The expression "with a certain probability" is more convient to the previous version that considers an arbitrary probability $1-\eta$ in my opinion. I would replace "with a certain probability" to "almost everywhere" or more precisely $D^{2m}$- or $D^{\otimes 2 m}$- almost everywhere.., as in the review of Reviewer FJNX.
>
> We have fixed this on Line 169 and also at the end of Definition 3.1.
>
> ---
>
> > Line 169: $z\in D$ does not hold as $D$ is not a set. Either $z\in \mathcal{Z}$ or $z\sim D$ .
>
> We have fixed this now [Lines 175,177].
>
> ---
>
> > Line 326: The $\eta$ almost $\beta$ bounded difference is mentioned. However, it is only defined in the appendix, so a remark (even between parentheses) referring to the appendix is necessary.
>
> This was a mistake and we have corrected it. We wanted to write a.s. support stability [Line 335].
>
> ---
>
> > Line 400: It is mentioned that "we need bound on just the square of the expectation of each term." By the Cauchy-Schwarz inequality, we have $E[L_S.L_g] \leq \sqrt{E[L_S^2]E[L_g^2]}$ . Hence, I think it is rather "the expectation of the square (or the second moment) of each term".
>
> The reviewer is correct. We have fixed this now [Line 411].
>
> ---
>
> > In line 710, I appreciate adding the "NN with identity function" precision to facilitate the interpreation of the base polynomial. However, as I read it "a fully connected NN defined in ..., i.e. NN with the identity function", this implies that the neural network defined in section D.1.1. has an identity activation function which is not the case. Also, the formulation makes it look as if the base polynomial is defined for an NN with an identity activation function, whereas it can be defined as the output for any fully connected NN if we assume it has the identity activation, while its weights are kept.
>
> The reviewer is right. As suggested by the reviewer, we have now made a distinction between the NN as defined and a  *different version* of the NN where the ReLU activation at each node is replaced by the identity activation. The base polynomial is then defined over this version.
>
> ---
>
> > In my previous review, in the comment about the sources for inspiration in the proofs, I was meaning for example "[Reference R, Theorem T]" instead of just "[Refernce R]". It is not necessary but its role is to locate the result quickly in the mentioned references. I apologize for any lack of clarity.
>
> We had not understood the request earlier. It makes perfect sense and we have now given the specific theorem references throughout the paper.
>
> ---
> ## Questions
> > In line 251, in assumption S1, it is written: "$L_l$ -LPL w.r.t. $\text{supp}D$ and "$K_l$ -LPS w.r.t. $D$ ". Is there a difference between saying "w.r.t. $D$ " or $\text{supp}D$ ?
>
> This is a typing error, thank you for pointing it out. $K_l$ will only be w.r.t $S$ as it only depends on training set. We have fixed this now [Line 260].
>
> ---
>
> > Is it possible to develop more on the advantages on the product of norms in Proposition 5.3, in terms of interpretability ?
>
> Several works in the literature have commented on the connection between spectral norms like the ones we have presented in Prop 5.3 (e.g. [Bartlett et al.](https://arxiv.org/pdf/1706.08498.pdf), [Lin et al.](https://proceedings.neurips.cc/paper_files/paper/2021/file/4ffb0d2ba92f664c2281970110a2e071-Paper.pdf)) and some works also use spectral regularizers to improve generalization ([Yoshida et al.](https://arxiv.org/pdf/1705.10941.pdf)). However, there is no clear answer in the literature for how we can interpret such norms. The general feeling is that a low product of norms implies a distribution that is ''easier'' to learn and a high product of norms implies a distribution that is ''harder'' to learn. That may be one way of interpreting the message of works such as [Rahaman et al.](https://arxiv.org/pdf/1806.08734.pdf) . However, it is very hard to make such a statement mathematically rigorous. But this question of the reviewer is very thought provoking and could lead to interesting lines of research.

---

### Review · Reviewer_2QBs · 2023-10-27

**Summary Of Contributions:**

This paper considers the notion of stability of learning algorithms, which usually implies good generalization bounds. It introduces the concept of *almost sure support stability*, and relates this concept to generalization bounds using a modified McDiarmid inequality.
It then shows that the SGD algorithm satisfies such support stability, with the bounds depending both on the data distribution and the weights encountered during the SGD training. Finally, the authors prove that for fully connected ReLU networks, those bounds are controlled by a spectral norm hypothesis on the weights of the network.

**Audience:**

Yes

**Broader Impact Concerns:**

N.A

**Claims And Evidence:**

Yes

**Requested Changes:**

In my opinion, the paper requires a much larger discussion on the scaling of the three constants $L_S$, $K_S$ or $L_g$; the ReLU network example still consists in a norm supremum over all possible trajectories of SGD, and thus is still very delicate to bound. Even a toy model for which those constants can be easily bounded could already be a good addition to the paper.

At the very least, a discussion on what those constants imply on the training procedure is in order. For example, the authors state that "if the model can’t fit the training set properly then these values will be high and reflect a bad generalization"; why would a high training loss imply high Lipschitz constants ? Regarding section 5.1.1, under which conditions can we expect that $L_S$ and $L_g$ are close ?

Minor remarks:
- I found the theorem statements quite heavy to read; a lot of the preambles are very similar, and should be broken down into separate assumptions. Less inline math should also be used.
- the notion of "symmetric in distribution" is never defined

**Strengths And Weaknesses:**

This paper manages to obtain good generalization bounds for SGD under a fairly realistic setting; in particular, it encompasses the multi-epoch regime, and the allowed decay of the learning rate in terms of both $m$ and $t$ seems much closer to what is actually used in practice.

However, I am unconvinced regarding the applicability of the obtained results. Compared to similar data-dependent bounds such as the ones in (Hardt et al. 2016) or (Kuzborskij and Lampert 2018), the constants $L_S$, $K_S$ and $L_g$ have a very complicated dependence in the data distribution and the specifics of the SGD dynamics (which can be very hard to analyze). Further, it seems to me like those bounds can increase arbitrarily in $m$ (as in the random label example), and the conditions under which $L_S$, $K_S$ or $L_g$ are bounded seem very hard to verify compared to previous work.

Further, all of section 5.3 consists in unsubstantiated claims about extensions of the results to other settings, without much theoretical footing; either those are easy adaptations and the theorem statements could be adapted to include those changes, or there are significant challenges that should be underlined.

---

> ### Author Response · Authors · 2023-11-10
> **Rebuttal by Authors**
>
> We would like to thank the reviewers for their close reading and their detailed comments. We have made an attempt to address these comments as fully as possible. However, in the process our rebuttal document has become very long and will exceed the 5000 word limit that Openreview places on posts. So, we have placed the entire rebuttal at the end of the updated pdf. We request the reviewers to find our responses there.
>
> In the main paper we have marked the edited text in 2 colors. Red indicates that the changes are important and add some extra information/value to the paper. Blue indicates the changes are mainly to increase readability, and they do not affect the paper's claims.
> We request the reviewers to go through them and respond and help us further improve this work.
>
> Thanks.

---

> > ### Comment · Reviewer_2QBs · 2023-11-13
> > **Post-rebuttal comment**
> >
> > I have read both the rebuttal and the new version of the paper. I welcome the additional clarity changes; however I am still unconvinced by the added example to illustrate the applicability of the theorems. Indeed, the example shows that the test Lipschitz constant $L_g$ can be better bounded when $W_1$ ranges across the space of bounded vectors. However, as the training goes, I would expect $W_1$ to align with the structure of $x$ and hence to be fully concentrated in its first $D$ coordinates as well. This would imply that the upper bounds over $\Omega$ and $\mathrm{supp}(D)$ actually coincide...
> >
> > This discrepancy is a good example of the (in my opinion) flaw in the results : the need to bound the Lipschitz constants *over all possible training trajectories* makes it extremely difficult to provide examples where there is a significant improvement over the previous results.
> >
> > New minor remarks:
> > - l. 158-159 : "where [S] denotes" and "where [m] represents" should be merged
> > - l. 169 : "uniformly" -> "uniform"
> > - l. 169 : $\forall z \in D$ -> $\forall z \in  \mathcal{Z}$
> > - l. 181 : "misleaing" -> "misleading"
> > - l. 315 : $c \log(2)$ -> $c_1 \log(2)$ ?

---

> > > ### Author Response · Authors · 2023-11-21
> > > **Response to Post-rebuttal comment**
> > >
> > > Thank you for your response. We have updated the draft and marked recent changes with magenta. We are posting replies to your comment here and have also maintained it in the latest draft.
> > >
> > > > I have read both the rebuttal and the new version of the paper. I welcome the additional clarity changes; however I am still unconvinced by the added example to illustrate the applicability of the theorems. Indeed, the example shows that the test Lipschitz constant $L_g$ can be better bounded when $W_1$ ranges across the space of bounded vectors. However, as the training goes, I would expect $W_1$ to align with the structure of $x$ and hence to be fully concentrated in its first $D$ coordinates as well. This would imply that the upper bounds over $\Omega$ and $\text{supp}(D)$ actually coincide...
> > > >
> > > > This discrepancy is a good example of the (in my opinion) flaw in the results : the need to bound the Lipschitz constants over all possible training trajectories makes it extremely difficult to provide examples where there is a significant improvement over the previous results.
> > >
> > > The reviewer's observation that as the training proceeds $W_1$ aligns with the subspace of the training points is correct. And, in fact, this is the crux of our work. The reviewer appears to claim that previous works can easily adopt this idea into their own frameworks but, to the best of our understanding, it is not trivial to do so and requires the machinery we have carefully developed for this purpose.
> > >
> > > Having said that, we appreciate the spirit of the reviewer's comment. This comment pushed us to further develop the material in Appendix C and we now believe that we have demonstrated the kind of concrete applicability that the reviewer has been concerned about from the beginning. In the first rebuttal we had added an alternate analysis that gives a generalization bound based on the square root of the second moment of the Lipschitz constants encountered during training $\sqrt{\mbox{E}_{r}\left[L\_{S,I_0}^2\right]}$. In the current revision of the paper we show in Appendix C.2 how a theorem of (Bottou et. al., 2018) can be used to bound this quantity in a variety of settings. This result is presented as Corollary C.4. In Appendix C.3 we consider a specific two-class classification problem learned by a two-layer NN and show a generalization bound for this case in terms of the parameters of the problem (Theorem C.5).
> > >
> > > We hope that the reviewer will agree that the framework of Appendix C.2 is quite useful, as illustrated by the concrete example in Sec C.3.  We feel that this is indeed the missing piece of the puzzle and we are grateful to the reviewer for pushing us to develop it. The ease with which our alternate analysis connected with the theory developed by (Bottou et. al. 2018) was exciting for us as we felt that it shows that we have now reached a way of looking at the problem that has appropriate resonances in the works of others.
> > >
> > > > New minor remarks:
> > >
> > > We have fixed the minor remarks. Thank you for the careful look over our paper.

---

### Decision · Action_Editor_9YNd · 2023-12-12

**Recommendation:** Accept as is

**Comment:**

The reviewers initially highlighted a considerable number of issues around clarity, definitions, and the overall framing of problems, which the authors acknowledged could be improved and addressed in their revisions. Specific technical concerns, such as the handling of "constants", the formulation of theorems, and the interpretation of results, were largely validated by the authors' detailed responses and revisions. Two of the reviewers were of the opinion that the reviews addressed their concerns.

Some areas where authors disagreed or clarified concerns from reviewers were misconceptions or misunderstandings about their framework, especially regarding its applicability and the interpretation of certain constants and results as mentioned above. They disagreed with the notion that their approach was overly complicated or less applicable than previous works, providing additional examples and discussions to demonstrate its relevance and utility.

There are some aspects of the rebuttal that might still leave reviewers disappointed. Primarily, reviewers may be left wanting more concrete demonstrations or simpler explanations, especially in terms of the practical applicability of the theoretical framework.
The complexity of the constants and the bounds, although clarified, are still be a point of contention for reviewers looking for more straightforward or intuitive results.

In summary, the authors have made a concerted effort to address the reviewers' concerns, focusing on clarifying their theoretical framework, refining definitions and theorems, and providing additional context and examples to demonstrate the applicability of their work. While some issue remain that may ultimately make the work less impactful, I believe it is suitable for publication.

**Audience:**

This paper will be of broad interest to the TMLR readership. It pertains to the generalization performance of deep learning, relating statistical quantities of a trained neural network to properties of the trajectory taken by the optimization algorithm used to train it. The work connects its theory to practice by empirically evaluating the hypotheses used to establish the result.

**Claims And Evidence:**

This is a theoretical paper, establishing a rigorous connection between the generalization of SGD and the norms of gradients encountered during training. This is achieved through a new notion of stability. The reviewers did not find any fault with the arguments presented. The authors present an empirical study of neural networks in order to evaluate whether the hypotheses of their theorems are met. They present evidence in support of this.

---

> ### Author Response · Authors · 2023-12-13
> **Thank you**
>
> Dear Prof Roy,
>
> Thank you for taking this paper to a timely conclusion. We appreciate your contributing your time to this work in the middle of all your other commitments. We are also deeply grateful to the reviewers. Their insightful comments and their uncompromising stance really pushed us to go deeper and helped us discover layers in this work that we had not thought of when we submitted. We thank them and wish them the very best.
>
> Thanks again and best wishes.
>
> The authors